# VISION HOPFIELD MEMORY NETWORKS

## ABSTRACT

Recent vision and multimodal foundation backbones, such as Transformer families and state-space models like Mamba, have achieved remarkable progress, enabling unified modeling across images, text, and beyond. Despite their empirical success, these architectures remain far from the computational principles of the human brain, often demanding enormous amounts of training data while offering limited interpretability. In this work, we propose the Vision Hopfield Memory Network (V-HMN), a brain-inspired foundation backbone that integrates hierarchical memory mechanisms with iterative refinement updates. Specifically, V-HMN incorporates local Hopfield modules that provide associative memory dynamics at the image patch level, global Hopfield modules that function as episodic memory for contextual modulation, and a predictive-coding–inspired refinement rule for iterative error correction. By organizing these memory-based modules hierarchically, V-HMN captures both local and global dynamics in a unified framework. Memory retrieval exposes the relationship between inputs and stored patterns, making decisions more interpretable, while the reuse of stored patterns improves data efficiency. This brain-inspired design therefore enhances interpretability and data efficiency beyond existing self-attention- or state-space–based approaches. We conducted extensive experiments on public computer vision benchmarks, and V-HMN achieved competitive results against widely adopted backbone architectures, while offering better interpretability, higher data efficiency, and stronger biological plausibility. These findings highlight the potential of V-HMN to serve as a next-generation vision foundation model, while also providing a generalizable blueprint for multimodal backbones in domains such as text and audio, thereby bridging brain-inspired computation with large-scale machine learning.

## 1 INTRODUCTION

The foundation models in computer vision have experienced significant changes in recent years. Starting with AlexNet (Krizhevsky et al., 2012) and its revolutionary performance, convolutional neural networks (CNNs) attracted the attention of researchers, leading to the design of advanced architectures such as VGG (Simonyan & Zisserman, 2015) and ResNet (He et al., 2016). Subsequently, the evolution of network architectures in natural language processing, particularly the Transformer (Vaswani et al., 2017), gave rise to its vision counterpart, the Vision Transformer (ViT) (Dosovitskiy et al., 2021), which achieved promising results on computer vision benchmarks. More importantly, the Transformer family established a strong connection between natural language processing and computer vision, unifying the modeling paradigm for both vision and language. Building on this trend, a variety of alternative architectures have been proposed, such as MLP-Mixer (Tolstikhin et al., 2021), MetaFormer (Yu et al., 2024), and more recently, state-space models like Vision Mamba (Vim) (Zhu et al., 2024), further enriching the landscape of foundation models for vision.

However, these models do not fundamentally address some of the long-standing challenges in deep learning. Specifically, they are not data-efficient and usually require large-scale datasets for training. Moreover, they lack biological plausibility, as their learning mechanisms differ substantially from how the human brain operates. In terms of data efficiency, current models rely heavily on extensive supervised training and large annotated datasets, which limit their applicability in domains where data collection is time-consuming or even infeasible. In contrast, humans are able to

learn robust concepts from very limited examples, pinpointing the gap between artificial and natural learning. As for biological plausibility, deep learning architectures and optimization methods are largely engineered for computational convenience rather than grounded in neuroscience. For example, backpropagation with gradient descent has no clear biological counterpart, and the conventional feedforward architecture overlooks key properties of the human brain, such as associative memory retrieval (Ramsauer et al., 2020) and predictive error correction (Rao & Ballard, 1999; Friston, 2005).

To deal with these challenges, we propose a new vision foundation model, named **Vision Hopfield Memory Network (V-HMN)**. V-HMN departs from conventional feedforward or self-attention-only designs by augmenting each block with *content-addressable associative memory*[1]. Concretely, V-HMN employs two complementary memory paths: (i) a *local window memory* that collects $k \times k$ neighborhoods and performs Hopfield-style retrieval to denoise and complete local patterns; and (ii) a *global template path* that forms a scene-level query via global pooling, retrieves a global prototype from memory, and injects it back into all tokens as a context prior. Both memory paths update features through an iterative refinement step with a learnable strength parameter. This mechanism can be viewed as a lightweight form of predictive-coding dynamics, where representations are gradually corrected toward memory-predicted patterns. In this way, the network gains an error-corrective feedback process that is absent in conventional feedforward models.

## 2 RELATED WORK

We now briefly review related works, including existing vision foundation models, associative memory and modern Hopfield networks, and brain-inspired predictive coding frameworks.

### 2.1 VISION FOUNDATION BACKBONES

Early advances in vision backbones were driven by convolutional neural networks (CNNs), such as AlexNet, VGG, and ResNet. While these models achieved remarkable progress, recent research has shifted toward alternative token-mixing paradigms. The ViT showed that a pure Transformer on image patches can rival CNNs at scale (Dosovitskiy et al., 2021), and hierarchical designs like Swin Transformer (Swin-ViT) (Liu et al., 2021) improved efficiency through shifted local windows. In addition, a line of hybrid architectures attempts to combine the complementary strengths of CNNs and Transformers. Representative examples include ConViT (d'Ascoli et al., 2021), which introduces soft convolutional inductive biases into attention layers, and CoaT (Xu et al., 2021), which integrates co-scale convolution with multi-head attention for better local–global trade-offs. Such hybrids highlight the ongoing interest in balancing locality and global context within a unified backbone. Beyond attention, MLP-based models (e.g., MLP-Mixer (Tolstikhin et al., 2021)) and MetaFormer[2] frameworks (Yu et al., 2022; 2024) demonstrated that different mixers can operate within a similar architectural scaffold. Most recently, state-space models (SSMs) have emerged as competitive backbones. S4 introduced structured SSMs for long sequences (Gu et al., 2021), and Mamba extended this idea with selective input-dependent dynamics (Gu & Dao, 2023). Vision-specific variants such as Vim (Zhu et al., 2024) and VMamba (Liu et al., 2024) show promising results with linear-time complexity. While these advances broaden the landscape of vision backbones, they typically lack explicit, interpretable memory mechanisms.

A recent attempt to address this limitation is the Associative Transformer (AiT) (Sun et al., 2025), which introduces a global workspace layer where memory slots are written via bottleneck attention and retrieved through Hopfield-style dynamics to refine token embeddings. While AiT shows the potential of integrating associative memory into Transformers, the most related work to ours, it remains fundamentally Transformer-based: memory serves as an auxiliary component, and heavy self-attention is still required for token interactions. By contrast, V-HMN is a memory-centric backbone in which local and global Hopfield modules fully replace self-attention as the token-mixing mech-

---

[1] In Hopfield networks, content-addressable memory refers to the ability to retrieve a stored pattern by filling in missing or noisy parts of the input.

[2] Throughout our experiments, we follow common practice and instantiate MetaFormer using PoolFormer, which is the default implementation adopted in prior work.

anism. This design makes V-HMN lighter, more interpretable, and closer to biologically inspired refinement principles, positioning memory not as an add-on but as the core of the backbone itself.

## 2.2 Associative memory and modern Hopfield networks

Modern Hopfield Networks (MHNs) revisit content-addressable memory with continuous states and an energy function that yields exponentially large storage and single-step retrieval in theory (Ramsauer et al., 2020). This line has been integrated into practical deep architectures via a differentiable Hopfield layer and applied beyond vision (e.g., retrieval, pooling, representation learning) (Ramsauer et al., 2020; Fürst et al., 2022). Recent works refine robustness and capacity, and study retrieval dynamics under modern settings (Wu et al., 2024; Hu et al., 2024). Compared to self-attention, Hopfield-style modules maintain a *persistent* memory bank with explicit slots (prototypes), enabling interpretable slot activations and prototype–token alignments. Our V-HMN leverages this by combining a local window memory (prototype completion/denoising) and a global template path (scene-level prior), both trained end-to-end.

Beyond Hopfield-style associative retrieval, a broader line of work has explored *prototype memories* and *external memory banks* as mechanisms for improving data efficiency. Early metric-based few-shot learning methods such as matching networks (Vinyals et al., 2016) and prototypical networks (Snell et al., 2017) explicitly store class-level prototypes in an embedding space and perform inference via metric retrieval, showing that maintaining persistent prototypes can substantially improve generalization under limited supervision. Subsequent extensions, including relation networks (Sung et al., 2018) and MetaOptNet (Lee et al., 2019), further refine prototype-based retrieval by learning similarity functions or optimizing embedding geometry for more reliable few-shot generalization. Prototype memory has also been explored in generative modeling. The approach of Li et al. (2022) maintains a learned bank of visual prototypes and retrieves them via attention to guide synthesis from only a handful of examples. Similarly, Qiao et al. (2022) introduce a prototype-conditioned generative mechanism in which retrieved prototypes act as structural priors that stabilize low-data generation. Both lines of work demonstrate that reusing persistent prototypes can effectively expand the statistical support of limited datasets and improve sample efficiency in generative scenarios.

## 2.3 Predictive-coding−inspired iterative refinement

Predictive coding (PC) is a long-standing theory in neuroscience that frames perception as iterative error minimization between predictions and sensory input (Rao & Ballard, 1999; Friston, 2005). In computational neuroscience, PC networks (PCNs) have been studied as a biologically plausible alternative to backpropagation (Whittington & Bogacz, 2019), while early deep learning variants such as PredNet applied the idea to video prediction with hierarchical recurrent modules (Lotter et al., 2017). More recent works have attempted to formalize PCNs in machine learning, drawing connections to variational inference, energy-based models, and equilibrium propagation (Millidge et al., 2022; van Zwol et al., 2024). Despite their theoretical appeal, PC-inspired models remain limited to small-scale or domain-specific settings, in part due to optimization difficulties and inefficiency in large-scale vision tasks. In this work, we do not implement full PC inference; instead, our blocks perform a lightweight, learnable refinement toward memory-predicted prototypes. This provides an interpretable, error-corrective step that is inspired by PC principles, connecting HMN-style memory refinement with a brain-inspired narrative, while keeping the backbone simple and scalable.

## 3 Methodology

In this section, we give an overview of the overall architecture of V-HMN, outlining how images are processed from patch embeddings to memory-based refinement blocks and finally to classification.

## 3.1 Overall Architecture

The V-HMN is designed as a memory-centric vision backbone. An input image is first projected into a sequence of image patch tokens. These tokens are then processed by a stack of HMN blocks, each integrating local and global Hopfield memory modules. Finally, the sequence is aggregated by attention pooling, and fed into a linear classifier. In contrast to convolutional backbones (ResNet

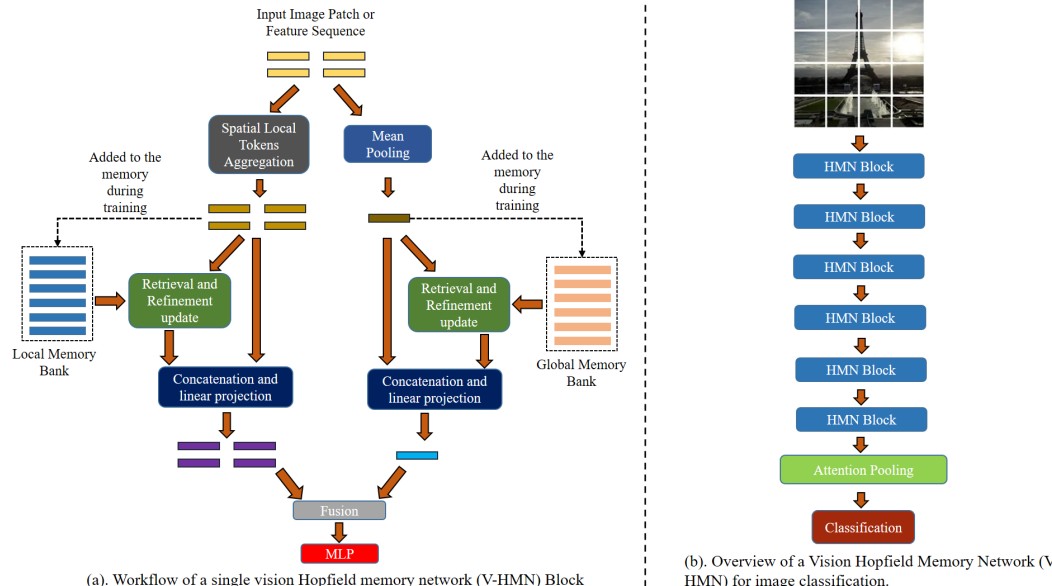

Figure 1: Overview of V-HMN. (a) Each HMN block refines features through local and global Hopfield memory retrieval, rather than convolution or self-attention. (b) A deep backbone is constructed by stacking HMN blocks, with attention pooling and a linear head for image classification.

(He et al., 2016)), self-attention-based designs (ViT (Dosovitskiy et al., 2021)), or state-space models (Vim (Zhu et al., 2024)), V-HMN replaces the underlying token-mixing operation with explicit associative memory retrieval. This makes memory refinement—not convolution, self-attention, or recurrence—the core building block of the backbone.

## 3.2 LOCAL AND GLOBAL HOPFIELD MEMORY MODULES

Each HMN block contains two complementary modules that together capture fine-grained local structure and holistic global context.

**Local memory.** For each token, its $k \times k$ spatial neighborhood is unfolded and projected into a latent space. Hopfield retrieval is then performed against a class-balanced memory bank that stores prototype features. The retrieved prototype refines the local representation by stabilizing noisy features and completing partial patterns, before being projected back to the embedding dimension and added to the residual stream. This mechanism parallels the role of convolutions or windowed attention, but operates through explicit prototype-based priors.

**Global memory.** To provide scene-level context, the global branch first mean-pools all tokens to form a query vector. This query interacts with a global memory bank through Hopfield retrieval, producing a prototype that captures global semantics. The result is broadcast to all tokens and integrated with the local path.

The overall workflow of a V-HMN block and its integration into the classification backbone are illustrated in Figure 1. Figure 1 (a) details a single block: the input token sequence first branches into (i) a *local memory* path that aggregates $k \times k$ spatial neighborhoods, performs associative retrieval with iterative refinement, and concatenates the refined and initial representations; and (ii) a *global memory* path that mean-pools all tokens to form a scene-level query, applies Hopfield-based retrieval and iterative refinement, and concatenates the refined and initial queries. The two paths are then fused by summation, and the fused representation is passed through a lightweight two-layer MLP, forming the output of the MHN block. In particular, the associative memory retrieval can be formalized as follows. Given a representation $z \in \mathbb{R}^D$ and a memory bank $M \in \mathbb{R}^{K \times D}$ with $K$ prototype slots, we first $\ell_2$-normalize both $z$ and the memory slots to obtain cosine similarities:

$$\hat{z} = \frac{z}{\|z\|_2}, \qquad \hat{M}_j = \frac{M_j}{\|M_j\|_2}, \; j = 1, \dots, K.$$

Retrieval weights and the retrieved prototype are then computed by

$$\alpha = \text{softmax}\left(\sqrt{D}\, \hat{z}\, \hat{M}^\top\right) \in \mathbb{R}^K, \qquad m = \sum_{j=1}^K \alpha_j\, M_j \in \mathbb{R}^D,$$

where $M_j$ is the $j$-th *unnormalized* memory slot, $\alpha$ are normalized weights ($\sum_j \alpha_j = 1$), and $m$ is the retrieved prototype. The additional scaling factor $\sqrt{D}$ is introduced, because cosine similarities $\hat{z}\hat{M}_j^\top$ have variance in the order of $1/D$: multiplying by $\sqrt{D}$ restores the logits to approximately unit variance [3], preventing the softmax distribution from becoming overly flat and yielding sharper, more discriminative retrieval weights.

Figure 1 (b) shows the full classification backbone formed by stacking multiple HMN blocks in depth, followed by attention pooling and a linear classifier. Specifically, attention pooling performs a weighted combination over all tokens to produce a single representation, which can be defined as:

$$\alpha = \text{softmax}(H\, W_{\text{att}}) \in \mathbb{R}^N, \qquad v = \sum_{i=1}^N \alpha_i\, H_i \in \mathbb{R}^D,$$

where $H \in \mathbb{R}^{N \times D}$ denotes the token representations after the final block, $W_{\text{att}} \in \mathbb{R}^{D \times 1}$ is a learnable scoring vector that assigns importance weights to tokens, $\alpha$ are the normalized attention weights ($\sum_i \alpha_i = 1$), and $v$ is the pooled representation fed into the classifier.

During training, both local and global modules maintain their own class-balanced memory banks. Unlike parametric weights, these banks are explicitly written with real sample embeddings at each block: the local bank stores projected patch-neighborhood features, while the global bank stores pooled scene-level features. Each bank is organized as a per-class ring buffer with fixed capacity, ensuring that all classes are allocated equal slots. As training proceeds, new embeddings replace the oldest ones within each class, yielding a continually refreshed and balanced set of prototypes. During inference, the banks are frozen and no longer updated, so retrieval always operates on stable prototypes that persist across tasks.

### 3.3 ITERATIVE REFINEMENT

The central operation in both local and global modules is an iterative refinement rule. Given a current representation $z$ and a retrieved prototype $m$, the update is

$$z^{(t+1)} = z^{(t)} + \beta\,(m - z^{(t)}), \tag{1}$$

where $\beta$ is a learnable update strength and $t$ denotes the refinement step. This mechanism can be viewed as a *predictive-coding–inspired* update: the prototype $m$ provides a memory-based prediction, while the residual $(m - z^{(t)})$ acts as a prediction error that gradually corrects the current representation, in line with the associative memory's role of filling in missing or noisy information. In contrast to full predictive coding networks that maintain explicit error units and multi-layer recurrent inference, V-HMN adopts a lightweight refinement loop where only a few steps are sufficient in practice. This refinement design leverages *persistent, content-addressable prototypes* that are shared across samples, and it yields two key benefits: (i) improved data efficiency, as stored prototypes provide reusable priors; and (ii) enhanced interpretability, since prototype activations directly expose the memory patterns supporting each decision.

### 4 EXPERIMENTS

We now report on our experiments on four public image classification benchmarks. To save space, implementation details are given in the appendix.

### 4.1 DATASETS

We evaluated our model on four widely used image classification benchmarks: CIFAR-10 (Krizhevsky & Hinton, 2009), CIFAR-100 (Krizhevsky & Hinton, 2009), SVHN (Netzer et al., 2011), and Fashion-MNIST (Xiao et al., 2017). CIFAR-10 and CIFAR-100 each consist of 60,000

---

[3]A detailed proof is provided in Appendix A.1.

Table 1: Data efficiency of V-HMN on CIFAR-10, CIFAR-100, and Fashion-MNIST with different fractions of labeled training samples. Reported values are top-1 test accuracy (%).

| Fraction of training data | CIFAR-10 | CIFAR-100 | Fashion-MNIST |
|---|---|---|---|
| 10% | 79.88 | 42.32 | 89.19 |
| 30% | 88.37 | 62.18 | 90.88 |
| 50% | 90.66 | 68.88 | 91.52 |

Table 2: Comparison of data efficiency across models with 10% and 30% labeled training data on CIFAR-10, CIFAR-100, and Fashion-MNIST. Reported values are top-1 test accuracy (%). Baselines include ViT (Dosovitskiy et al., 2021), Swin-ViT (Liu et al., 2021), MLP-Mixer (Tolstikhin et al., 2021), MetaFormer (Yu et al., 2024), Vim (Zhu et al., 2024), and AiT (Sun et al., 2025).

| Model | CIFAR-10 | CIFAR-100 | Fashion-MNIST |
|---|---|---|---|
| **10% training data** | | | |
| ViT | 72.73 | 40.48 | 87.17 |
| Swin-ViT | 70.37 | 35.25 | 88.42 |
| MLP-Mixer | 76.14 | 41.94 | 87.16 |
| MetaFormer | 49.92 | 19.39 | 85.02 |
| Vim | 66.58 | 37.28 | 86.38 |
| Assoc. Transformer | 67.89 | 35.84 | 84.33 |
| **V-HMN (ours)** | **79.88** | **42.32** | **89.19** |
| **30% training data** | | | |
| ViT | 83.94 | 57.40 | 89.71 |
| Swin-ViT | 80.89 | 51.77 | 90.34 |
| MLP-Mixer | 85.53 | 56.22 | 89.41 |
| MetaFormer | 60.86 | 37.47 | 88.78 |
| Vim | 79.58 | 49.11 | 88.77 |
| Assoc. Transformer | 80.69 | 54.59 | 87.62 |
| **V-HMN (ours)** | **88.37** | **62.18** | **90.88** |

color images with a resolution of $32 \times 32$. Each dataset is split into 50,000 training samples and 10,000 test samples, covering 10 and 100 object categories, respectively. SVHN is a large-scale street view digit recognition dataset with more than 600,000 labeled images at $32 \times 32$ resolution. Compared to CIFAR, it is more challenging due to cluttered backgrounds, overlapping digits, and significant intra-class variation. Fashion-MNIST provides 70,000 grayscale images of size $28 \times 28$ from 10 categories of clothing and accessories. It was introduced as a modern alternative to the classic MNIST digits, with richer texture and shape variations. For large-scale evaluation, we additionally conduct experiments on ImageNet-1k (Deng et al., 2009), which contains 1.28M training images and 50K validation images across 1,000 classes.

## 4.2 ABLATION STUDIES

**Data Efficiency.** We study the data efficiency of V-HMN under varying fractions of labeled training data. Table 1 shows that accuracy improves steadily as the proportion of labeled data increases: even with only 10% of the training set, V-HMN achieves competitive performance, while scaling to 30% and 50% further closes the gap to the full-data regime. This indicates that the associative memory mechanism provides strong inductive biases that reduce dependence on large-scale annotation.

We further benchmark V-HMN against standard vision backbones under 10% and 30% labeled data (Table 2). Across CIFAR-10, CIFAR-100, and Fashion-MNIST, V-HMN consistently outperforms widely used architectures such as ViT, Swin-ViT, MLP-Mixer, and MetaFormer, as well as more recent models like Vim. Importantly, V-HMN also surpasses the AiT, which integrates memory slots into a Transformer backbone. This highlights the benefit of our design: rather than appending memory to an existing architecture, V-HMN makes associative memory the core computational primitive. The resulting prototype-based refinement provides stronger gains, especially in low-data settings, where stored prototypes act as reusable priors and compensate for scarce supervision.

Table 3: Ablation study on the number of refinement iterations in V-HMN. Top-1 test accuracy (%) are reported.

| Iterations | CIFAR-10 | CIFAR-100 | Fashion-MNIST |
|---|---|---|---|
| 0 | 92.26 | 75.31 | 92.12 |
| 1 | 93.48 | 76.20 | 92.64 |
| 2 | 93.99 | 75.95 | 92.91 |
| 3 | 93.72 | 75.75 | 92.76 |

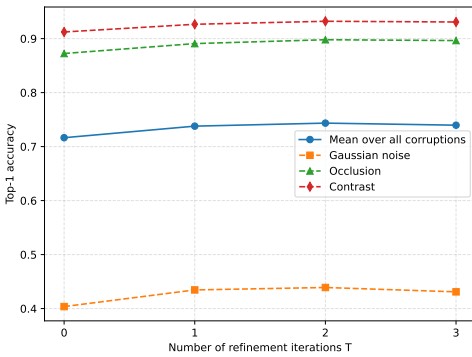

Figure 2: Effect of refinement iteration on CIFAR-10 robustness across Gaussian noise, occlusion, and contrast corruptions.

**Iterative Refinement.** Table 3 summarizes the effect of varying the number of refinement iterations $t$. Here, $t = 0$ disables the associative refinement loop entirely: the local and global branches still compute their feedforward projections, but no Hopfield retrieval or error-correction update (Eq. 1) is applied. The memory banks remain allocated during training to keep the parameter count identical, but they are not read during inference.

Across all datasets, introducing even a single refinement step ($t = 1$) yields consistent improvements (e.g., CIFAR-10: 92.26% → 93.48%; CIFAR-100: 75.31% → 76.20%). This confirms that the associative update provides meaningful benefits, although the underlying feedforward pathway is already strong. Performance peaks at $t = 2$, while deeper unrolling offers no additional gain and can slightly degrade accuracy due to over-correction. This trend is consistent with predictive-coding models of cortical processing, where a small number of recurrent error-correction steps typically suffices to explain the input and further unrolling mainly increases computational cost (Rao & Ballard, 1999; Friston, 2005).

In addition, we conduct robustness experiments to assess whether additional refinement iterations offer benefits beyond a single update. We evaluate CIFAR-10 models trained with different numbers of refinement steps under several corruptions: Gaussian noise (standard deviations 0.05, 0.10, 0.20, 0.30), random square occlusion (areas 0.05, 0.10, 0.20), and contrast scaling (factors 0.5, 0.75, 1.25, 1.5). As shown in Figure 2, we report both the mean top-1 accuracy across all corruptions and the per-corruption accuracy for each refinement depth. Averaged over all corruptions, accuracy increases from 71.65% at $t = 0$ to 73.79% at $t = 1$ and 74.35% at $t = 2$. The gains are most pronounced for occlusion and contrast: occlusion accuracy improves from 87.24% ($t = 0$) to 89.08% ($t = 1$) and 89.79% ($t = 2$), with similar improvements for contrast scaling. Overall, these results demonstrate that the predictive-coding–inspired refinement yields measurable and consistent robustness gains. Unless otherwise specified, we fix the number of iterations to $t = 1$ in all other experiments for a balanced trade-off between accuracy and efficiency.

**Sizes of Memory.** Table 4 summarizes the effect of varying the sizes of the local and global memory banks. Performance does not grow monotonically with larger memories; instead, the best results arise from moderate capacities, approximately 2500 slots for local memory and 1000 slots for global memory. This pattern complements the iteration ablation: because the model performs only a small number of refinement steps, what matters most is having a well-curated, high-quality prototype set that can provide targeted corrections, rather than an excessively large bank.

When the memories are too small, they under-cover the feature space and limit the corrective power of each refinement step. When they are excessively large, the bank becomes redundant and introduces retrieval noise, slightly reducing accuracy and weakening the influence of each update. Together with the iteration ablation, these results indicate that associative retrieval functions as a lightweight, prototype-based prior on top of a strong feedforward backbone. While not required for basic recognition (as seen from the small drop at $t = 0$), the learned memories and 1–2 refinement iterations consistently improve robustness and data efficiency once an appropriate prototype set is established.

Table 4: Ablation study on the effect of local and global memory sizes in V-HMN. Top-1 test accuracy (%) are reported.

| Local memory size | | | | Global memory size | | | |
|---|---|---|---|---|---|---|---|
| Size | CIFAR-10 | CIFAR-100 | Fashion-MNIST | Size | CIFAR-10 | CIFAR-100 | Fashion-MNIST |
| 1500 | 93.12 | 75.78 | 92.25 | 500 | 93.11 | 76.17 | 92.04 |
| 2500 | 93.48 | 76.20 | 92.64 | 1000 | 93.48 | 76.20 | 92.64 |
| 3500 | 93.17 | 75.99 | 92.28 | 1500 | 92.97 | 75.32 | 92.36 |
| 4500 | 92.95 | 75.69 | 92.06 | 2000 | 93.03 | 76.02 | 92.44 |

Table 5: Comparison of V-HMN with baseline models on CIFAR-10, CIFAR-100, SVHN, and Fashion-MNIST. Top-1 test accuracy (%) are reported. Baselines include ViT (Dosovitskiy et al., 2021), Swin-ViT (Liu et al., 2021), MLP-Mixer (Tolstikhin et al., 2021), MetaFormer (Yu et al., 2024), Vim (Zhu et al., 2024), and AiT (Sun et al., 2025).

| Model | CIFAR-10 | CIFAR-100 | SVHN | FashionMNIST | Params (M) |
|---|---|---|---|---|---|
| ViT | 91.05 | 71.53 | 95.32 | 91.11 | 7.16 |
| Swin-ViT | 89.26 | 66.91 | 96.02 | 91.68 | 6.92 |
| MLP-Mixer | 91.93 | 73.15 | 96.80 | 90.46 | 8.71 |
| MetaFormer | 82.68 | 59.99 | 90.93 | 90.57 | 6.99 |
| Vim | 77.31 | 43.34 | 85.02 | 87.01 | 7.01 |
| AiT | 92.30 | 73.20 | 95.86 | 91.40 | 7.15 |
| V-HMN (ours) | **93.48** | **76.20** | **96.90** | **92.64** | 7.60 |

## 4.3 MAIN RESULTS

Table 5 reports the comparison of V-HMN with a wide range of mainstream vision foundation backbones, including Transformer-based models (ViT, Swin-ViT), MLP-based architectures (MLP-Mixer, MetaFormer), state-space models (Vim), and the recently proposed AiT. Despite having a comparable parameter scale, V-HMN consistently achieves the best performance across all benchmarks, reaching 93.48% on CIFAR-10, 76.20% on CIFAR-100, 96.90% on SVHN, and 92.64% on Fashion-MNIST.

We attribute these improvements to two key design choices. First, the incorporation of *local and global associative memories* enables the model to retrieve and integrate prototypical patterns, effectively supplementing limited supervision with reusable priors. Second, the *iterative refinement mechanism* provides a lightweight error-corrective process that gradually aligns representations with memory-predicted prototypes, thereby enhancing the robustness. Compared with standard feed-forward Transformers or purely sequential state-space models, V-HMN benefits from persistent, content-addressable prototypes that capture both local details and global context, yielding stronger generalization under comparable model capacity.

A key observation is that V-HMN surpasses the AiT (Sun et al., 2025) across all datasets, despite having nearly identical parameter counts. While AiT integrates associative memory *within* a Transformer layer, its token mixing remains attention-centric. In contrast, V-HMN is *memory-centric*: local and global Hopfield modules replace self-attention as the mixing mechanism, memories are persistent and class-balanced (written with real sample embeddings during training and frozen at inference), and representations are updated through a predictive-coding–inspired refinement loop. These results suggest that making associative memory the *primary* computational primitive, rather than an auxiliary component to Transformer, provides better data efficiency and accuracy under comparable model capacity.

To further evaluate the scalability of V-HMN, we conduct experiments on ImageNet-1k and report results in Table 6. Without any architectural specialization for large-scale training, V-HMN achieves accuracy comparable to widely used CNN and Transformer backbones of similar parameter counts. This finding is encouraging given that V-HMN is a newly proposed architecture: unlike mature baselines that have benefited from years of iterative optimization and carefully engineered design heuristics, V-HMN has not undergone extensive hyperparameter tuning or additional engineering effort. Although V-HMN demonstrates strong data efficiency on smaller benchmarks, our ImageNet

Table 6: Comparison on ImageNet-1k. All results are Top-1 accuracy (%).

| Method | Image Size | #Params | Top-1 Acc. |
|---|---|---|---|
| ResNet-18 (He et al., 2016) | 224 | 12M | 70.6 |
| ResNet-34 (He et al., 2016) | 224 | 22M | 75.5 |
| ResNet-50 (He et al., 2016) | 224 | 26M | 79.8 |
| ViT-B/16 (Dosovitskiy et al., 2021) | 384 | 86M | 77.9 |
| ViT-L/16 (Dosovitskiy et al., 2021) | 384 | 307M | 76.5 |
| MLP-Mixer-B/16 (Tolstikhin et al., 2021) | 224 | 59M | 76.4 |
| Swin-Mixer-T/D24 (Liu et al., 2021) | 256 | 20M | 79.4 |
| PVT-Small (Wang et al., 2021) | 224 | 25M | 79.8 |
| V-HMN | 224 | 88M | 79.8 |

experiment serves a different purpose. Rather than competing with heavily optimized architectures such as RegNetY (Radosavovic et al., 2020), DeiT (Touvron et al., 2021), Swin-ViT (Liu et al., 2021), or Vim (Gu & Dao, 2023), the goal is to assess whether the model's core inductive bias remains viable at large scale. Despite minimal tuning, V-HMN remains competitive on ImageNet, indicating that the framework is not only biologically inspired but also inherently interpretable and sufficiently general to operate effectively at high-resolution, large-dataset regimes. Moreover, V-HMN preserves a transparent computational structure, namely explicit associative-memory retrieval coupled with predictive-coding–inspired iterative refinement, that is difficult to maintain in more complex or heavily engineered architectures. This transparency further enables interpretability of the decision process, since predictions can be explained by identifying which stored patterns are retrieved and refined.

### 4.4 VISUALIZATION

To better understand the behavior of V-HMN, we visualize the retrieved prototypes from both local and global memory banks. For each test image, we first identify the most informative patch based on the attention pooling weights (highlighted by the red box), and then retrieve its nearest prototypes from the corresponding memory banks. To clearly illustrate the retrieved regions, we adopt a patch size of $8 \times 8$ during visualization.

Figure 3 shows representative results across four categories: automobile, ship, dog, and deer. Several interesting observations can be made. First, the *local memory* retrieves highly similar local structures from other images of the same class. For example, in the automobile and ship cases, the retrieved prototypes consistently align to similar positions, highlighting that local memory captures part-level correspondences across samples. This demonstrates that local memory is capable of associating fine-grained visual parts across different samples, thereby enhancing spatial interpretability. Second, the *global memory* retrieves holistic prototypes that provide complementary, scene-level priors. For instance, as shown in the dog and deer cases, the global retrieval captures diverse poses and viewpoints of the same class, which supply additional information to complete or refine the local representation. This behavior is consistent with the role of associative memory, which not only recalls similar exemplars but also provides missing context to stabilize predictions.

Together, these results highlight that the two memory paths confer complementary benefits: local memory aligns semantically corresponding parts across samples, while global memory furnishes broader priors that help disambiguate incomplete or noisy inputs. Such explicit retrieval and refinement significantly improve the interpretability of V-HMN, as one can directly inspect which prototypes drive the model's decision process.

## 5 SUMMARY AND OUTLOOK

In this work, we introduced **V-HMN**, a brain-inspired vision backbone that augments each block with local and global Hopfield-style memory. Through associative retrieval and predictive-coding–inspired refinement, V-HMN moves beyond purely feedforward or self-attention architectures and places memory at the center of feature integration. This yields two key benefits: *data efficiency*, by reusing stored prototypes as inductive priors, and *interpretability*, as retrieved prototypes explicitly

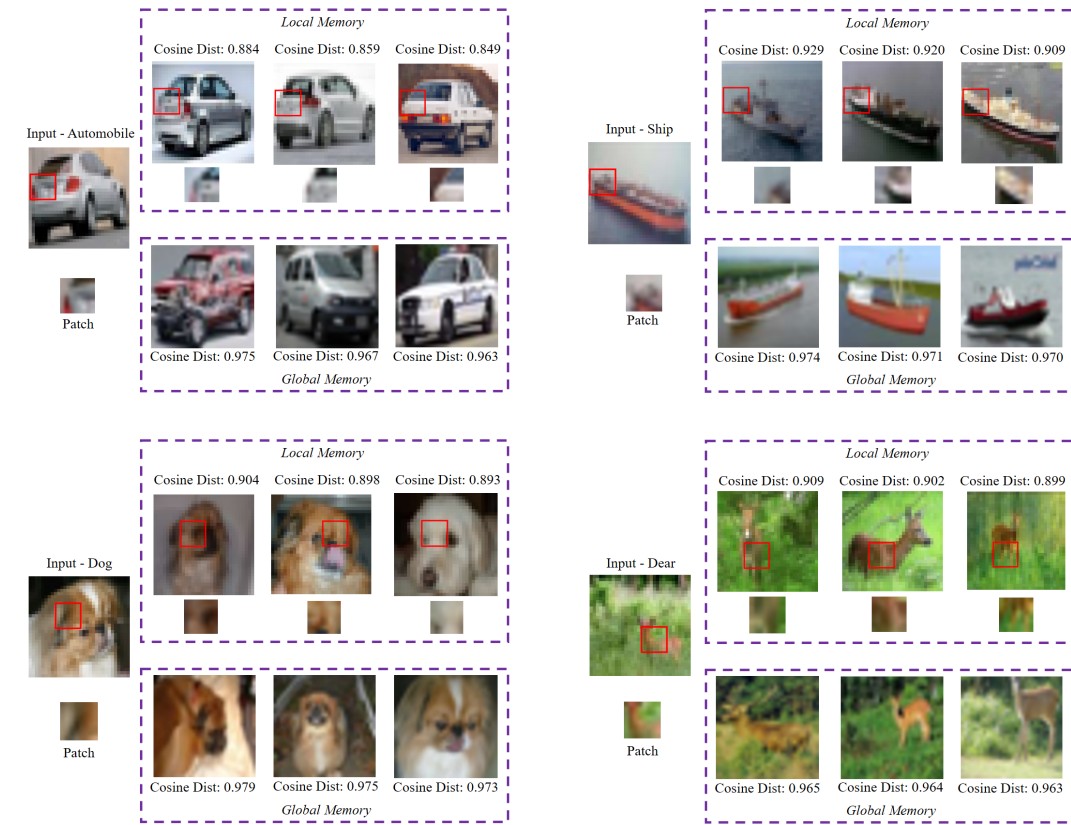

Figure 3: Visualization of retrieved prototypes from local and global memory.

reveal the patterns influencing each prediction. V-HMN outperforms vision backbones of comparable scale across CIFAR, SVHN, and Fashion-MNIST, and furthermore demonstrates that its inductive bias scales reliably to ImageNet, achieving competitive performance without large-scale architectural tuning and hyper-parameter search.

Our study highlights the promise of memory-centric backbones as a competitive and scalable alternative to mainstream vision foundation models. By grounding representation learning in explicit associative memory and lightweight error-corrective refinement, V-HMN bridges biologically inspired computation with large-scale machine learning. This memory-driven principle is not limited to classification; rather, it suggests a broader applicability across a range of vision tasks. Beyond classification, memory-centric architectures such as V-HMN are inherently well suited for a broader range of vision tasks. Because the model operates through explicit associative retrieval, it can naturally support retrieval and metric-learning tasks, where matching against stored prototypes is a core operation rather than an auxiliary mechanism. For example, the prototype-based memory organization further aligns with few-shot learning and generative modeling (Li et al., 2022), enabling rapid adaptation by recalling class-level or instance-level representations without requiring extensive optimization. In dense prediction settings such as segmentation and detection, the combination of local spatial memory and global scene-level recall provides an additional advantage: local memory supports fine-grained spatial refinement, while global memory offers holistic contextual priors that stabilize predictions across regions. This hierarchical memory interaction can serve as an effective inductive bias for tasks requiring multi-scale reasoning and structured outputs. Overall, these applications highlight how memory-based systems are not merely an architectural variant, but introduce a distinct computational principle, namly explicit pattern storage, retrieval, and refinement, that may generalize across diverse vision tasks.

We believe these directions open the door to more interpretable, data-efficient, and biologically inspired architectures.

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

# A APPENDIX

## A.1 VARIANCE OF COSINE SIMILARITY

Let $\hat{q}, \hat{k} \in \mathbb{R}^D$ be independent random unit vectors. We first compute the second moment of $\hat{q}$. Write $\hat{q} = (\hat{q}_1, \ldots, \hat{q}_D)$. By coordinate symmetry, all coordinates satisfy $\mathbb{E}[\hat{q}_i^2] = v$. Since $\sum_{i=1}^{D} \hat{q}_i^2 = 1$, taking expectations gives

$$Dv = 1 \quad \Rightarrow \quad v = \tfrac{1}{D}.$$

For $i \neq j$, flipping the sign of the $i$-th coordinate (which preserves the distribution of a random unit vector) implies

$$\mathbb{E}[\hat{q}_i \hat{q}_j] = -\mathbb{E}[\hat{q}_i \hat{q}_j] \quad \Rightarrow \quad \mathbb{E}[\hat{q}_i \hat{q}_j] = 0.$$

Hence the off-diagonal entries vanish and the diagonal entries are all $1/D$, i.e.,

$$\mathbb{E}[\hat{q}\,\hat{q}^\top] = \tfrac{1}{D}I.$$

It follows that

$$\mathbb{E}[\hat{q}^\top \hat{k}] = \mathbb{E}_{\hat{q}}\big[\hat{q}^\top \, \mathbb{E}_{\hat{k}}[\hat{k}]\big] = 0.$$

For the second moment,

$$\mathbb{E}\big[(\hat{q}^\top \hat{k})^2\big] = \mathbb{E}_{\hat{k}}\Big[\hat{k}^\top \, \mathbb{E}_{\hat{q}}[\hat{q}\,\hat{q}^\top] \, \hat{k}\Big]$$
$$= \mathbb{E}_{\hat{k}}\Big[\hat{k}^\top \Big(\tfrac{1}{D}I\Big)\hat{k}\Big]$$
$$= \tfrac{1}{D} \, \mathbb{E}_{\hat{k}}[\|\hat{k}\|_2^2]$$
$$= \tfrac{1}{D}.$$

Therefore,

$$\mathrm{Var}(\hat{q}^\top \hat{k}) = \mathbb{E}\big[(\hat{q}^\top \hat{k})^2\big] - \big(\mathbb{E}[\hat{q}^\top \hat{k}]\big)^2 = \tfrac{1}{D}.$$

Concluding, the variance of the cosine similarity between two random unit vectors decays as $1/D$. Multiplying the similarity by $\sqrt{D}$ rescales the logits to approximately unit variance before the softmax.

## A.2 ITERATIVE REFINEMENT AS PREDICTIVE-CODING (PC) DYNAMICS

The central operation in both local and global modules is an iterative refinement rule. Given a current representation $z^{(t)} \in \mathbb{R}^D$ and a memory bank $M \in \mathbb{R}^{K \times D}$ with rows $M_j \in \mathbb{R}^D$, we first perform Hopfield-style retrieval as described in Section 3.2:

$$\hat{z}^{(t)} = \frac{z^{(t)}}{\|z^{(t)}\|_2}, \quad \hat{M}_j = \frac{M_j}{\|M_j\|_2},$$
$$\alpha_j^{(t)} = \mathrm{softmax}_j\big(\sqrt{D}\,\hat{z}^{(t)}\hat{M}_j^\top\big),$$
$$m^{(t)} = \sum_{j=1}^{K} \alpha_j^{(t)} M_j, \tag{2}$$

where $M_j$ denotes the $j$-th memory slot, and $m^{(t)}$ is the retrieved memory. We then update $z^{(t)}$ via

$$z^{(t+1)} = z^{(t)} + \beta\big(m^{(t)} - z^{(t)}\big), \tag{3}$$

where $\beta$ is a learnable update strength.

This rule can be interpreted as a simple predictive-coding (PC) dynamics. Define the local *prediction error*

$$\varepsilon^{(t)} := m^{(t)} - z^{(t)}.$$

Then, Eq. 3 becomes

$$z^{(t+1)} = z^{(t)} + \beta\,\varepsilon^{(t)},$$

which is a discrete gradient-descent step on the squared prediction-error energy

$$\mathcal{F}(z) \;=\; \tfrac{1}{2}\, \|z - m(z)\|_2^2.$$

If we treat $m(z)$ as fixed with respect to $z$ during one update step, then

$$\nabla_z \mathcal{F}(z) \;\approx\; z - m(z) \;=\; -\varepsilon^{(t)},$$

and Eq. 3 coincides with

$$z^{(t+1)} \;\approx\; z^{(t)} - \beta \nabla_z \mathcal{F}(z^{(t)}).$$

That is, the Hopfield module produces a memory-based prediction $m^{(t)}$ of the code $z^{(t)}$, the residual $\varepsilon^{(t)}$ acts as a prediction-error signal, and the representation is iteratively corrected, so as to minimize a local prediction-error energy.

This mirrors the core mechanism of hierarchical predictive coding (Rao & Ballard, 1999; Friston, 2005; Whittington & Bogacz, 2019): higher-level causes generate a prediction of lower-level activity, explicit error units encode their difference, and latent states are updated by gradient descent on a prediction-error or free-energy objective. In V-HMN, the role of the generative model is played by the associative memory: memory slots $M_j$ act as prototypical causes, the similarity scores $\alpha_j^{(t)}$ approximate a posterior over these causes given $z^{(t)}$, and the retrieved prototype $m^{(t)} = \mathbb{E}[M_j \mid z^{(t)}]$ provides the top-down prediction. The refinement dynamics of Eq. 3 thus implements a lightweight PC update in which (i) the *feedforward* path computes similarities and posterior weights from the current features to the memory slots, and (ii) the *feedback* path injects the memory-based prediction back into the features through the prediction-error signal $(m^{(t)} - z^{(t)})$. In practice, we find that one or two iterations are sufficient to obtain robust improvements while keeping computation efficient.

### A.3 HIERARCHICAL LOCAL–GLOBAL MEMORY AND INVARIANCES

V-HMN is organized as a stack of HMN blocks, each equipped with its own local and global memory banks. The hierarchical structure arises because each block processes and stores representations at its own depth: the prototypes learned at lower layers are grounded in early, fine-grained features, while deeper layers operate on progressively transformed and more semantically organized representations produced by the preceding blocks. As a result, lower layers learn prototypes of local edge- and texture-like patterns, whereas higher layers learn more abstract object- and scene-level prototypes.

**Layerwise interaction of local and global memories.** At every layer, the local HMN operates on overlapping neighborhoods of the token grid to retrieve and refine local patterns, while the global HMN pools information across all tokens to retrieve a scene-level prototype that is broadcast back to the entire layer. Importantly, each layer's memory banks only interact with the representations produced at that same depth; deeper layers never directly access shallow-layer features. This architectural separation induces a genuine hierarchy of prototypes. Lower layers store and reinstate fine-grained edge-, texture-, and patch-level patterns, whereas higher layers operate on increasingly abstract representations passed up from previous blocks. The retrieved global prototype at each layer provides a contextual prior that guides local refinement, enabling ambiguous or partially occluded patches to be interpreted in a way consistent with the overall scene. Through this layerwise progression, V-HMN builds increasingly holistic and context-aware representations without relying on explicit spatial downsampling.

**What invariances the hierarchy provides.** The hierarchical local–global design induces several useful invariances, but not all invariances observed in practice come from the architecture alone.

- **Local tolerance to small perturbations.** Because local neighborhoods are unfolded with overlap, a small translation or deformation of a feature (e.g., shifting an edge by one pixel) changes which tokens contribute to a neighborhood but often leaves its nearest local prototype unchanged. The local Hopfield retrieval therefore tends to map slightly perturbed patches back to the same or a nearby prototype, providing robustness to small local jitters and noise.

- **Contextual invariance via global prototypes.** The global HMN sees a pooled summary of the entire token grid. As a result, global prototypes are largely insensitive to where an object appears within the image, acting more like a translation-tolerant scene or object code. When the global prototype is broadcast back to all tokens, it stabilizes local representations against clutter or partial occlusion: different arrangements of the same global configuration are attracted to similar global memory slots.

- **Increasing invariance with depth.** As representations become progressively more abstract across successive HMN blocks, prototypes in deeper local and global memory banks become less sensitive to fine-grained pixel-level details and more sensitive to object- and class-level structure. This yields a degree of scale and translation tolerance at higher layers, analogous to the progression observed along the ventral visual stream.

**What is handled by augmentation, preprocessing, and pooling.** Several important invariances are primarily provided by standard deep-learning components rather than by the memory hierarchy itself. Random crops, horizontal flips, and optional AutoAugment are the main sources of robustness to large translations, flips, and complex photometric distortions. The patch embedding and final attention pooling contribute additional translation invariance by making the classifier depend mostly on aggregated token statistics rather than exact pixel coordinates. We do not build in explicit rotation or scale-equivariant structure; robustness to such transformations arises empirically from data augmentation and from the generic effects of depth and pooling, not from a specialised architectural mechanism.

In summary, the hierarchical arrangement of local and global memory banks provides a structured inductive bias: lower layers learn local prototypes that are robust to small perturbations; higher layers and global memories learn more holistic prototypes that are tolerant to object location and clutter. This interacts synergistically with standard augmentation and pooling to produce the overall invariance profile observed in our experiments.

### A.4    WHY V-HMN IS DATA-EFFICIENT

The empirical results in Tables 1 and 2 show that V-HMN maintains strong performance even when only a small fraction of the training labels is available. For instance, with just 10% of the labeled data, V-HMN achieves 79.88% on CIFAR-10, 42.32% on CIFAR-100, and 89.19% on Fashion-MNIST (Table 1), and it consistently outperforms ViT, Swin-ViT, MLP-Mixer, MetaFormer, Vim, and the AiT under both 10% and 30% training data across all three benchmarks (Table 2). We now briefly analyze why V-HMN is more data-efficient than these baselines.

**Prototype-based nonparametric prior.** Both the local and global Hopfield modules retrieve representations from explicit memory banks whose slots are populated with real latent embeddings during training. These slots act as prototypes that approximate class-conditional manifolds in feature space. Once a prototype is stored, future inputs can leverage it via retrieval even if supervision is limited, providing a nonparametric prior that complements the parametric backbone. In low-data regimes, this prototype reuse compensates for limited gradient-based fitting and reduces overfitting to the small labeled set. The monotonic improvements from 10% to 30% to 50% labeled data in Table 1 reflect this behavior: as more labeled examples are observed, memory banks become better populated and the same prototypes can be reused across many inputs.

**Local–global inductive bias.** V-HMN enforces a structured inductive bias by combining (i) local Hopfield dynamics on unfolded $k \times k$ neighborhoods and (ii) a global Hopfield module operating on scene-level aggregates. Local memories specialize to recurring edge- and texture-like patches, while global memories capture higher-level scene and object prototypes. This hierarchical organization constrains the effective hypothesis class: new images are explained in terms of reusing and recombining a finite set of learned local and global prototypes, rather than learning fresh features from scratch for every configuration. This reduces the amount of labeled data needed to reach a given accuracy, which is consistent with the stronger gains of V-HMN in the 10% and 30% settings compared to the full-data regime in Table 2.

**Iterative refinement focuses capacity on hard examples.** As discussed in Section A.2, the refinement rule implements a predictive-coding (PC) update, where representations are iteratively corrected to reduce local prediction-error energy. In practice, this means that model capacity is concentrated on those regions of feature space where the memory-based predictions and current representations disagree. When data are scarce, this mechanism helps stabilize learning: easy examples quickly align with their nearest prototypes and require little further adjustment, while scarce or atypical examples receive more refinement steps. The ablation in Table 3 shows that removing refinement ($t = 0$) substantially hurts performance and that one or two refinement iterations yield the best trade-off between accuracy and computation. Together with the memory-based priors above, this targeted refinement explains why V-HMN achieves higher accuracies than standard backbones and the AiT under limited supervision (Table 2), despite using a comparable number of parameters.

### A.5 BIOLOGICAL PLAUSIBILITY AND RELATION TO CORTICAL CIRCUITS

V-HMN is not intended as a faithful anatomical model of the primate visual system, but it is explicitly guided by two computational motifs that are widely discussed in systems neuroscience: (i) associative memory implemented by recurrently connected populations, and (ii) predictive-coding-like error-corrective refinement in cortical microcircuits. We now clarify where our design aligns with known hippocampal and cortical circuitry and where the connection is only analogical.

**Associative memory beyond the hippocampus.** Associative dynamics are often introduced via models of the hippocampal formation, where pattern separation in dentate gyrus and pattern completion in CA3 are thought to support episodic memory and spatial navigation. However, a large body of theoretical and experimental work suggests that related forms of autoassociative computation are also implemented in neocortical microcircuits, where recurrent collateral connections between pyramidal neurons can sustain stable activity patterns that function as long-term and short-term memories, perceptual representations, and decision states. In this broader view, associative memory is a *general* cortical computational motif, not something confined to the hippocampus. Our local Hopfield modules are designed to echo this idea: they implement content-addressable retrieval over local patch-level embeddings using modern Hopfield dynamics, such that frequently co-occurring visual features (e.g., edges, corners, textures) form structured prototype-like representations stored in the memory bank. These modules enable V-HMN to integrate local contextual information in a manner that is both interpretable and robust, supporting refinement steps that adjust features toward semantically consistent local patterns, even under perturbations.

**Global memory and hippocampal/entorhinal inspiration.** At the same time, the global Hopfield module and its class-balanced episodic memory bank are more directly inspired by hippocampal and entorhinal circuitry. The global query aggregates information across the entire image and retrieves a scene-level prototype from a separate memory bank, loosely analogous to how hippocampus and entorhinal cortex integrate inputs from many cortical areas into a sparse episodic code that can later be reinstated to bias cortical activity. The broadcast of the retrieved global prototype back to all tokens in a block is therefore reminiscent of hippocampo–cortical feedback that reinstates context or episodes, but at a highly abstract level and without any claim of anatomical fidelity.

**Relation to ventral visual stream processing.** In the brain, visual object recognition emerges along the ventral visual stream (V1, V2, V4, IT), with rich recurrent connectivity and local microcircuits, before hippocampal structures become involved in binding objects into episodic memories. V-HMN compresses some of these ideas into a single backbone: local HMN blocks can be viewed as abstracted cortical microcircuits that combine feedforward feature extraction with local associative retrieval and predictive-coding-style refinement, and the global HMN adds an additional, hippocampus-inspired contextual signal. We do not claim a one-to-one mapping between layers of V-HMN and specific cortical areas. Rather, our aim is to capture, within a practical vision model, the core computational principles of (i) distributed pattern reinstatement through associative memory and (ii) iterative interaction between top-down predictions and bottom-up features.

**Brain-inspired, not anatomically faithful.** Taken together, these design choices place V-HMN somewhere between conventional deep vision backbones and detailed biophysical models. Compared to standard feedforward or self-attention-only architectures, V-HMN moves closer to known

cortical and hippocampal computation by making associative memory and error-corrective dynamics first-class citizens in the backbone: memory banks correspond to explicit, inspectable prototypes rather than hidden weights; Hopfield retrieval implements pattern completion; and iterative refinement reduces local prediction errors. At the same time, we abstract away from many anatomical details (layer-specific connectivity, cell types, precise ventral-stream staging), and we do not claim that V-HMN is a literal model of the primate visual system. Instead, our goal is to move towards more biologically grounded computation than in current deep learning models, while remaining competitive and scalable on standard machine-learning benchmarks.

## A.6 Implementation Details

V-HMN is implemented in PyTorch. As for the experiments on small datasets (CIFAR, Fashion-MNIST, and SVHN), images are divided into non-overlapping $4 \times 4$ patches and embedded into tokens with a learnable positional encoding. The backbone consists of six V-HMN blocks, each equipped with a local Hopfield module operating on $3 \times 3$ spatial neighborhoods and a global Hopfield module operating on the mean-pooled query. The local and global modules maintain class-balanced memory banks with 2500 and 1000 slots, respectively. The embedding and latent dimensions are both set to 300, and the MLP expansion ratio is set to 2. Refinement updates are controlled by a learnable parameter $\beta$ initialized to 0.2, and unless otherwise specified, a single refinement iteration is used. We use the Adam optimizer (Kingma & Ba, 2014) with cosine learning-rate decay. The initial learning rate is 0.001, linearly warmed up during the first five epochs. Training is conducted for 400 epochs with a batch size of 256 and weight decay of $5 \times 10^{-5}$. For data augmentation, we adopt a strong augmentation including random cropping, horizontal flipping, AutoAugment (Cubuk et al., 2019), MixUp (Zhang et al., 2017), and CutMix (Yun et al., 2019). All reported results are based on this training setup unless otherwise noted.

As for the ImageNet experiments, we use a batch size of 1000 and a patch size of $14 \times 14$. The local and global memory banks contain 5000 and 3000 slots, respectively. We train the model for 310 epochs using the AdamW optimizer with a cosine learning-rate schedule. The embedding and latent dimensions are both set to 576, and the MLP expansion ratio is set to 4. The initial learning rate is 0.001 with linear warmup at the beginning of training. A weight decay of 0.03 is applied, and gradient clipping with a maximum norm of 0.1 is used. For data augmentation, we use RandAugment, MixUp, CutMix, and random erasing. We do not use stochastic depth, repeated augmentation, or Exponential Moving Average (EMA).

## A.7 Explorations on Spatial Window Size

Table 7: Ablation on spatial window size $k$. Accuracy (%) is reported on CIFAR-10, CIFAR-100, and Fashion-MNIST.

| Window size $k$ | CIFAR-10 | CIFAR-100 | Fashion-MNIST |
|---|---|---|---|
| 3 | 93.48 | 76.20 | 92.64 |
| 5 | 93.10 | 76.14 | 92.61 |
| 7 | 93.15 | 75.95 | 92.58 |

Table 7 reports the effect of varying the spatial window size $k$ for local memory retrieval. We observe that a smaller window ($k = 3$) yields the best results across datasets, while larger windows ($k = 5, 7$) lead to slightly lower accuracy. This suggests that restricting memory matching to a compact neighborhood is beneficial, as it enforces stronger locality priors and avoids interference from irrelevant patches. At the same time, the overall performance difference remains small, indicating that the model is robust to the choice of $k$.

## A.8 Explorations on $\beta$ Initialization

In this section, we study how the initialization of $\beta$ affects performance when the number of refinement steps is fixed to $t = 1$.

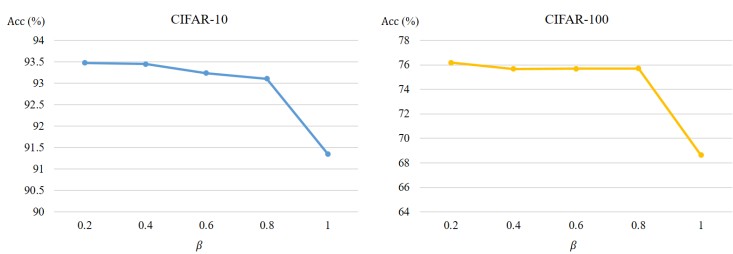

Figure 4: Effect of $\beta$ initialization on model accuracy with refinement iteration fixed to $t = 1$.

Figure 4 reports results on CIFAR-10 and CIFAR-100 for $\beta \in \{0.2, 0.4, 0.6, 0.8, 1.0\}$ (learned thereafter during training). Performance is relatively stable for small-to-moderate values, with the best accuracy obtained when initializing $\beta$ around $0.2$. Setting $\beta = 1.0$ consistently degrades accuracy on both datasets.

We hypothesize that the effect primarily reflects the strength of the refinement update. Very large initial values make the update overly aggressive—effectively pushing the representation too close to the retrieved prototype—while very small values under-utilize the memory-based correction. Intermediate initialization therefore provides a balanced step size that avoids both under-correction and over-writing, which explains the better performance observed for moderate $\beta$ values. This effect is important in the early stages of training, when the memory banks are still noisy and prototypes are less well formed; overly aggressive updates at this stage can amplify noise and hurt generalization.

Unless otherwise noted, we initialize $\beta$ to $0.2$ (and keep it learnable), which provides a robust starting point and yields the best or near-best results across datasets in this single-iteration setting.

Table 8: Performance under class-imbalance on CIFAR-10 and CIFAR-100. All results are top-1 accuracy (%). Baselines include ViT (Dosovitskiy et al., 2021), Swin-ViT (Liu et al., 2021), MLP-Mixer (Tolstikhin et al., 2021), MetaFormer (Yu et al., 2024), Vim (Zhu et al., 2024), and AiT (Sun et al., 2025).

| Method | Imbalanced CIFAR-10 | | Imbalanced CIFAR-100 | |
|---|---|---|---|---|
| | 50 | 100 | 50 | 100 |
| ViT | 72.43 | 64.46 | 42.66 | 37.30 |
| Swin-ViT | 65.79 | 59.56 | 38.25 | 34.37 |
| MLP-Mixer | 72.56 | 65.09 | 42.04 | 36.59 |
| Vim | 63.30 | 55.57 | 39.43 | 35.11 |
| AiT | 61.90 | 57.02 | 38.37 | 33.90 |
| Meta-Former | 56.78 | 50.77 | 30.70 | 26.85 |
| V-HMN | **76.33** | **71.71** | **46.82** | **41.43** |

## A.9 IMBALANCED SETTING

Following the long-tailed evaluation protocol in prior work such as Cao et al. (2019), Zhou et al. (2020), and Kang et al. (2020), we additionally assess V-HMN under class-imbalanced CIFAR-10/100. We construct imbalance ratios of 50 and 100, where an imbalance ratio of $r$ means that the number of samples in the most frequent class is $r$ times larger than that in the least frequent class. This setting not only makes the training data distribution highly skewed, but also induces imbalance in the learned prototype memories themselves, since minority classes contribute far fewer instances to the memory-update process.

Table 8 reports the results, in which 50 and 100 denote he imbalanced ratios. Across all imbalance ratios and datasets, V-HMN achieves the strongest robustness, substantially outperforming ViT, Swin-ViT, MLP-Mixer, Vim, AiT, and Meta-Former. We attribute this behavior to the associative-memory mechanism: unlike purely feedforward models, V-HMN maintains a set of learned prototypes that aggregate information across the entire training distribution. During inference, the refinement step retrieves class-relevant prototypes and corrects the latent representation toward them.

Even under severe imbalance, minority-class prototypes remain available in the memory bank and continue to provide stabilizing signals, mitigating representation drift and reducing majority-class dominance.

In summary, these results suggest that the memory-based associative refinement acts as an inherent regularizer in long-tailed regimes, allowing V-HMN to maintain accuracy where other architectures degrade more severely.

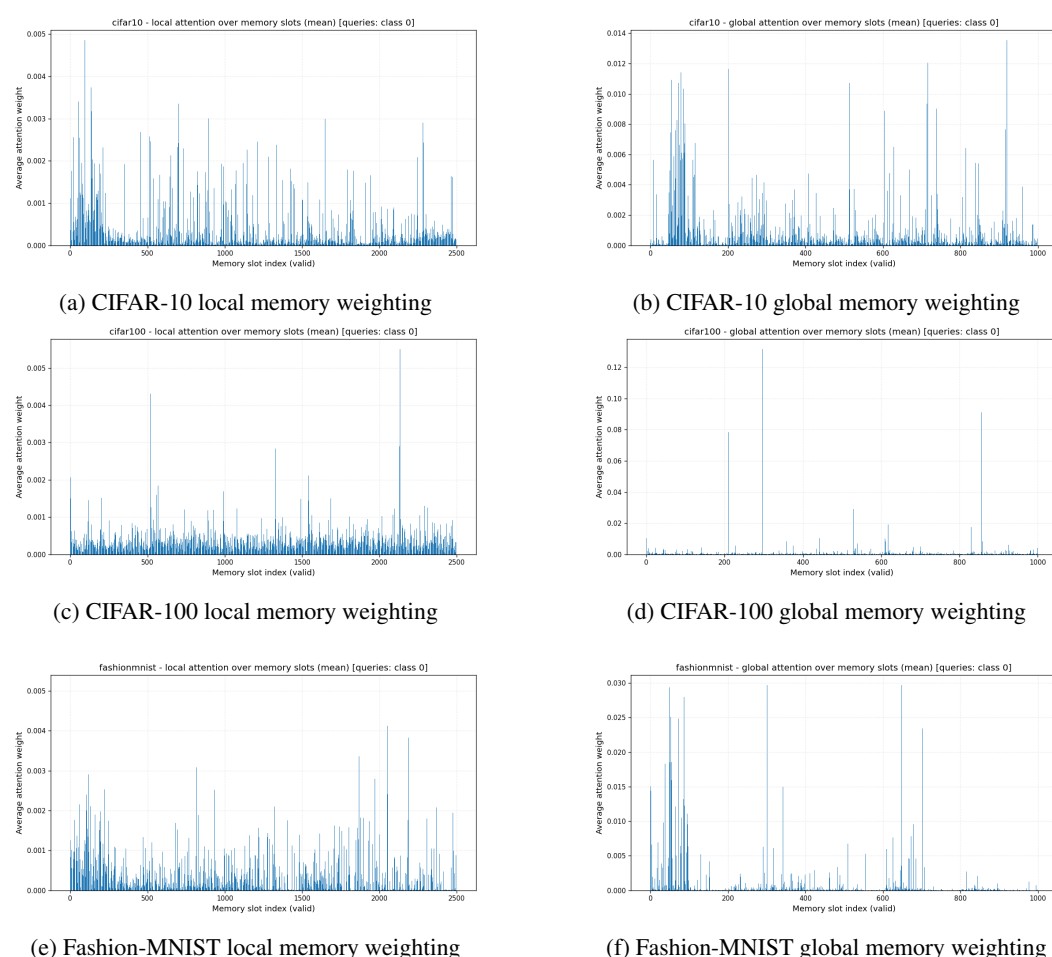

(a) CIFAR-10 local memory weighting

(b) CIFAR-10 global memory weighting

(c) CIFAR-100 local memory weighting

(d) CIFAR-100 global memory weighting

(e) Fashion-MNIST local memory weighting

(f) Fashion-MNIST global memory weighting

Figure 5: Visualization of the memory slot weightings of the last layer of the V-HMN

## A.10 Visualization of Memory Weights

We visualize how memory slots are weighted in the last layer of the model when inputs pass through it. We perform this visualization for CIFAR-10, CIFAR-100 and Fashion-MNIST. For each, we pass all inputs from the dataset's first class through the model, and average over the resulting memory slot weightings. The results are shown in Figure 5. For CIFAR-10 and Fashion-MNIST, the weights are highest on average in the first 10% of the memory slots, corresponding to the first dataset class. This shows that the refinement step refines the latent vectors towards prototypes of the same class. For CIFAR-100, no such patterns can be distinguished in the visualization, as the first class is only represented by the first 1% of memory slots, which is too little to be clearly visible.

Table 9: Top-1 and top-5 retrieval hit rate, in percent (%). Measured on CIFAR-10, CIFAR-100 and Fashion-MNIST.

|  | CIFAR-10 | CIFAR-100 | Fashion-MNIST |
|---|---|---|---|
| Local Top-1 Hit Rate | 30.87 | 1.92 | 35.73 |
| Local Top-5 Hit Rate | 61.38 | 6.73 | 73.44 |
| Global Top-1 Hit Rate | 36.32 | 7.82 | 68.80 |
| Global Top-5 Hit Rate | 81.43 | 24.19 | 96.24 |

## A.11 RETRIEVAL

We analyze how the model interacts with the memories by applying it to samples from CIFAR-10, CIFAR-100 and Fashion-MNIST and calculating how frequently the memory slots that are weighted the highest in the refinement steps come from the same class as the sample input. We consider both the top-1 hit rate, which measures how frequently the single highest-weighted memory slot is from the same class as the sample input, and the top-5 hit rate, which measures how frequently at least one of five highest-weighted memory slots is from the same class as the sample input. We measure these values in the last layer of the model. The results are shown in Table 9. If the retrieval was completely random the expected top-1 and top-5 hit rates would be 1% and 5%, respectively, for CIFAR-100, and 10% and 50%, respectively, for CIFAR-10 and Fashion-MNIST. In all cases, the hit rate is significantly higher than these values. This shows that the refinement process works as expected: The latent vectors are refined towards the stored memories of the same class.

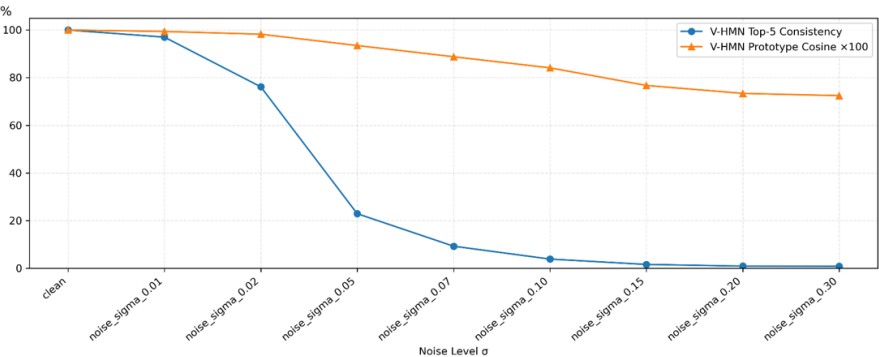

(a). Robustness of Hopfield memory network (V-HMN) under Gaussian noise

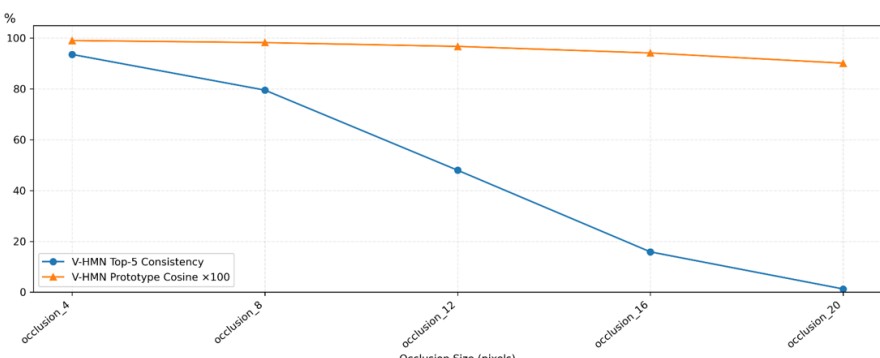

(b). Robustness of Hopfield memory network (V-HMN) under Occlusions.

Figure 6: Robustness analysis of V-HMN's associative memory under Gaussian noise and occlusion.

## A.12 ASSOCIATIVE MEMORY DYNAMICS IN V-HMN: ROBUSTNESS, INTERPRETABILITY, AND DATA EFFICIENCY

To evaluate the robustness of V-HMN's memory module as a *prototype memory*, we apply controlled perturbations to CIFAR-10 test images and examine how the retrieved prototypes change. We consider two perturbation types: (1) additive Gaussian noise with standard deviations $\sigma \in [0.01, 0.30]$, and (2) block occlusions with side lengths from 4 to 20 pixels (covering up to 39% of the input). To quantify how retrieval behaves under corruption, we evaluate two complementary metrics:

- **Top-5 Consistency (blue).** This metric evaluates the stability of *discrete* prototype indexing under perturbations. For each clean test image, we first record the set of its top-5 retrieved prototypes. After applying a perturbation, we obtain the corrupted image's top-1 retrieved prototype, and check whether this prototype is contained in the clean image's top-5 set. We repeat this for all test images and report the percentage of cases in which the corrupted top-1 prototype remains within the clean top-5 set. Higher values therefore indicate that the model maintains stable prototype choices even when the input is perturbed.

- **Prototype Cosine Similarity (orange).** This metric evaluates the *semantic* stability of prototype retrieval. For each test image, we compute the cosine similarity between: (i) the prototype vector retrieved as top-1 for the clean image, and (ii) the prototype vector retrieved as top-1 for its corrupted version. We average this cosine similarity across all test samples. High similarity indicates that even when the discrete index changes, the retrieved prototypes remain semantically similar (e.g., neighbors in prototype space).

In the occlusion experiments, the blue curve drops linearly and rapidly (falling below 20% at a $16 \times 16$ occlusion), indicating frequent index switching. In contrast, the orange curve remains highly robust, maintaining over 90% similarity even with $20 \times 20$ occlusions. This shows that V-HMN often switches to semantically adjacent "neighbor" prototypes, effectively performing pattern completion. The noise experiments exhibit a similar trend: while the blue metric is highly sensitive (dropping to $\sim 20\%$ at $\sigma = 0.05$), the orange metric retains around 70% similarity even under extreme noise ($\sigma = 0.30$), and approximately 90% at $\sigma = 0.07$. These results confirm that the associative retrieval mechanism provides strong denoising capabilities, mapping corrupted signals back toward the correct prototype manifold.

These findings demonstrate that V-HMN possesses strong semantic robustness, interpretability, and data efficiency. Across both noise and occlusion perturbations, the cosine similarity between pre- and post-perturbation prototypes remains remarkably stable, indicating that the model consistently maps corrupted inputs back toward nearby semantic centers in memory. This behavior reflects an effective many-to-one compression from high-dimensional pixel variations to a low-dimensional, structured prototype space, providing a principled explanation for V-HMN's data efficiency: diverse corrupted variants are absorbed into a small set of meaningful prototypes, reducing the need to observe every possible input configuration during training. Moreover, the model's ability to maintain high semantic similarity even when the discrete index changes underscores its interpretability—prototype switching occurs primarily among semantically adjacent neighbors, revealing a clear structure in the associative-memory dynamics. Finally, the smooth and predictable changes observed under increasing perturbation levels indicate stable and reliable retrieval behavior, without abrupt or erratic transitions. Together, these properties highlight the role of associative-memory refinement not merely as an architectural component, but as a robust inductive bias that enhances stability, generalization, and interpretability across challenging input conditions.

## STATEMENT ON LLM USAGE

We acknowledge the use of an LLM to refine grammar and improve readability of the manuscript. The research ideas, methodology, experiments, and conclusions were entirely conceived and validated by the authors.

