# OpenReview forum: "Vision Hopfield Memory Networks"
_ICLR.cc/2026/Conference — ICLR 2026 Conference Desk Rejected Submission_

### Official Review · Reviewer_qwXq · 2025-10-21

**Soundness:** 3
**Presentation:** 3
**Contribution:** 3
**Rating:** 4
**Confidence:** 5

**Summary:**

This paper introduces Vision Hopfield Memory Networks, a biologically inspired vision backbone that replaces standard token-mixing operations (like convolution or self-attention) with hierarchical Hopfield memory mechanisms. The model integrates two complementary modules: 1) Local Hopfield memory for patch-level associative retrieval and denoising, and 2) Global Hopfield memory for scene-level context and episodic modulation. Both modules interact through a predictive-coding–inspired iterative refinement rule, allowing representations to be gradually corrected based on stored prototypes.

**Strengths:**

* The motivation is very well defined and strong. I think this paper proposes an approach than can deal with fundamental problems in modern AI architectures such as equipping models with associative retrieval and predictive coding capabilities while both being neuro-inspired.
* Retrieved prototypes expose which stored patterns influence decisions which increases interpretability, a rare feature in vision backbones.
* Achieves strong results with as little as 10–30% of labeled data.
* The paper presents ablation studies.
* Results are strong even though on toy datasets.
* The paper presents qualitative results.

**Weaknesses:**

* While the predictive-coding analogy is conceptually appealing it remains lightweight where the connection could be deepened theoretically or experimentally.
* While the paper presents an appealing biologically inspired narrative, the underlying model is fairly simple, essentially a combination of local/global prototype retrieval and residual refinement. The connection to predictive coding and Hopfield dynamics is more analogical than formal, and the theoretical depth is limited. Nevertheless, the simplicity is also a strength: it demonstrates that interpretable, memory-based inductive biases can yield competitive results without architectural complexity even though on "toy" datasets.
* The experiments are too much toy. Empirical results on real world datasets or tasks could be done.
* The gains from increasing the number of refinement steps are minimal which can mean that the predictive coding strategy is not bringing much value.

Minor comments:
* Too many acronyms
* ViT acronym is repeated

**Questions:**

* Why didn't you use learnable projection layers in the HMN block to increase expressiveness?

**Details Of Ethics Concerns:**

I do not identify any significant ethical issues in this paper. The method operates on publicly available video datasets commonly used in the community, and there is no indication of privacy violations, harmful content generation, or misuse potential beyond standard concerns in visual understanding research. Therefore, I do not see any ethical concerns requiring further attention.

---

> ### Author Response · Authors · 2025-11-25
>
> We appreciate the reviewer’s insightful comment.
>
> # Weaknesses:
>
> **While the predictive-coding analogy is conceptually appealing it remains lightweight where the connection could be deepened theoretically or experimentally.**
>
> In the revision, we expand this connection by explicitly showing how our update rule follows the structure of a prediction-error–driven refinement step and by discussing its relation to energy-based formulations (see Sec A.2).
>
> At the same time, our goal is not to claim a full predictive-coding model, but to demonstrate that a minimal instantiation of the predictive-coding principle, iterative correction toward an internal prediction already provides measurable benefits. Thanks for prompting us to articulate this  more precisely.
>
> **While the paper presents an appealing biologically inspired narrative, the underlying model is fairly simple, essentially a combination of local/global prototype retrieval and residual refinement. The connection to predictive coding and Hopfield dynamics is more analogical than formal, and the theoretical depth is limited. Nevertheless, the simplicity is also a strength: it demonstrates that interpretable, memory-based inductive biases can yield competitive results without architectural complexity even though on "toy" datasets.**
>
> We agree with the reviewer’s observation that the core model is intentionally simple. Our goal is not to fully reconstruct the entire machinery of predictive coding or Hopfield dynamics, but to isolate the key inductive bias, namely, iterative error refinement over a  prototype memory, and examine whether this principle alone yields empirical benefits.
>
> We also agree that the connection to predictive coding and Hopfield networks is conceptual rather than a full formal derivation, and we clarified this positioning in the revision. At the same time, the connection is not superficial:
>
> -- Predictive coding: the residual used in our refinement step corresponds directly to the gradient of an energy function with respect to the representation (i.e., the prediction error term in classical predictive-coding updates).
>
> -- Hopfield-style memory: the global prototype retrieval step can be interpreted as an associative descent on a learned energy landscape.
>
> To enhance the connection to predictive coding and Hopfield dynamics, we added more contents in the revised paper, please refer to Sections A.2 and A.5.
>
> Furthermore, we have now included ImageNet-1k experiments. Please refer to lines 426-455 and Table 6.  These results demonstrate that the same lightweight memory-refinement mechanism remains effective at a larger scale, addressing concerns that our contributions apply only to “toy’’ datasets. V-HMN remains competitive on ImageNet, indicating that the framework is not only biologically inspired but also inherently interpretable and sufficiently general to operate effectively at high-resolution, large-dataset regimes. Plus, V-HMN preserves a transparent computational structure, namely, explicit associative-memory retrieval coupled with predictive-coding–inspired iterative refinement, that is difficult to maintain in more complex or heavily engineered architectures. This transparency further enables interpretability of the decision process, since predictions can be explained by identifying which stored patterns are retrieved and refined.
>
> **The experiments are too much toy. Empirical results on real world datasets or tasks could be done.**
>
> We have now included ImageNet-1k experiments. Please refer to lines 426-455, Table 6.
>
> **The gains from increasing the number of refinement steps are minimal which can mean that the predictive coding strategy is not bringing much value.**
>
> While the gains indeed saturate quickly, the first refinement step (t=0→1) already produces clear improvements across all datasets. This behavior is consistent with the predictive-coding–inspired view behind our design: most examples are well explained by a single or double error-correction–style update, and deeper unrolling mainly adds computation and potential over-correction, leading to diminishing returns, as commonly seen in iterative-inference frameworks.
> To further assess the value of our PC-inspired iterative refinement, we conducted additional robustness experiments on CIFAR-10 with Gaussian noise, occlusion, and contrast perturbations. As shown in Figure 2 of the revised manuscript, accuracy increases consistently as t increases (overall: 71.65→73.79→74.35), indicating that the refinement mechanism continues to provide meaningful benefits beyond a single step, particularly under corrupted inputs.
> Please refer to the new experiment  and analysis, which have been incorporated into the revised paper (Table3, Figure 2, lines 340-362).
>
> **Too many acronyms, ViT acronym is repeated**
>
> Thank you for the comment. We have revised the manuscript to improve clarity regarding acronym usage

---

> ### Author Response · Authors · 2025-11-25
>
> # Questions:
>
> **Why didn't you use learnable projection layers in the HMN block to increase expressiveness?**
>
> The HMN block already includes learnable projection layers. After the local and global memory retrieval, the fused features are passed through a learnable projection (implemented as a small MLP). This component was part of all experiments, but we did not emphasize it in the original submission, because our focus was on the memory mechanism. The revised version now clarifies this detail in lines 211-212 and Figure 1

---

> ### Comment · Reviewer_qwXq · 2025-11-25
> **Official Comment by Reviewer qwXq**
>
> The authors addressed my main concerns. I'm thus raising the score.

---

### Official Review · Reviewer_7PFy · 2025-10-30

**Soundness:** 4
**Presentation:** 3
**Contribution:** 3
**Rating:** 10
**Confidence:** 3

**Summary:**

This paper proposed a new type of vision foundational model using Hopfield associative memory mechanism as a building block. The basic idea is to have local patch neighborhoods encoded and stored in memory banks at the local level and to cover global characteristics another encodings derived from pooling across all the patches in a global memory bank. As new training images are added, the Hopfield update operations are performed in both banks to update the prototypical encodings stored in the respective memory bank. Each such Hopfield memory-laden block can be stacked as layers followed by an attentional pooling step to give the final encoded representation which is then fed to a classifier.

Experiments are conducted on a few benchmark datasets and comparison is performed to alternative vision foundational model paradigms based on ViT, VisionMamba and other architectures, and adequate ablation studies are done to expose the features. Overall, the model seems to have comparable number of parameters to the SOTA models, and the main advantage appears to be having biological motivation and fairly large improvement in classification performance with lesser training data.

**Strengths:**

The use of Hopfield retrieval circuits inside a vision foundational model is novel and very interesting. Effectively the operation is akin to local  and global clustering of patterns in encoding spaces derived originally from the image. The results are impressive as well.

**Weaknesses:**

It would be good to discuss the biological plausibility of the proposed mechanism, since the Hopfield networks are found later in the pathway within the Hippocampal system and take input from the entorhinal cortex by which time the image was already analyzed in the visual pathway and via the parahippocampal and perihinal cortex, the object location and identities have already been completed. So while this is a new and interesting vision model,  I am not convinced is reflective of what is going in the brain.

The hierarchical aspects of the model need further elaboration as is the rotational and other invariances in the model where the associative recall may not address those aspects.

**Questions:**

Hopfield networks are notorious for entering the metastable states. How does that effect the recognition capacity of the model. It would also be good to see if it is able to discriminate between very similar patterns using this mechanism.

If the number of parameters are about the same, is the main advantage of the model the improved accuracy (for which we should probably look at more datasets) or the amount of training data required. If more data is diluting the system due to the metastable states, then this may pose a limitation for this model which should be discussed.

Is the code for this being made available?

---

> ### Author Response · Authors · 2025-11-25
>
> We appreciate the reviewer’s highly positive assessment and thoughtful feedback.
>
> # Weaknesses:
>
> **It would be good to discuss the biological plausibility of the proposed mechanism, since the Hopfield networks are found later in the pathway within the Hippocampal system and take input from the entorhinal cortex by which time the image was already analyzed in the visual pathway and via the parahippocampal and perihinal cortex, the object location and identities have already been completed. So while this is a new and interesting vision model, I am not convinced is reflective of what is going in the brain.**
>
> We thank the reviewer’s insightful comment regarding biological plausibility. To avoid any misunderstanding, we have revised the manuscript to clearly state that V-HMN is not intended as an anatomical model of hippocampal circuitry. Instead, the architecture is guided by broad computational principles, associative retrieval and iterative error correction, that are observed both in hippocampal circuits and in recurrent neocortical microcircuits. In Section A.5, we now explicitly describe which aspects of the model draw inspiration from known neural computations and which aspects (e.g., layer correspondence, anatomical pathways) are not meant to map onto biological structures. We hope this clarification adequately addresses the reviewer’s concern.
>
> **The hierarchical aspects of the model need further elaboration as is the rotational and other invariances in the model where the associative recall may not address those aspects.**
>
> We have added a detailed explanation of the hierarchical aspects of V-HMN and how the model handles different forms of invariance.  Please kindly refer to Appendix Section A.3, where we describe:
>
> 1. how layerwise local–global memory interactions induce a hierarchy of prototypes,
>
> 2. which invariances arise from the architecture (e.g., local tolerance to small perturbations, contextual invariance via global prototypes), and
>
> 3. which invariances (e.g., large rotations or scale changes) are provided by data augmentation and generic deep-learning components rather than by associative recall itself.
>
> We hope this clarifies the inductive biases introduced by V-HMN and how they relate to invariance and hierarchical processing.

---

> > ### Author Response · Authors · 2025-11-25
> >
> > # Questions:
> >
> > **Hopfield networks are notorious for entering the metastable states. How does that effect the recognition capacity of the model. It would also be good to see if it is able to discriminate between very similar patterns using this mechanism.**
> >
> > To address the concern about metastable states and discrimination between very similar patterns, we performed a dedicated analysis of the prototype retrieval dynamics (Appendix A.12, Figure 6).
> >
> > For each clean test image and its corrupted counterparts (Gaussian noise or block occlusion), we measured (i) whether the corrupted sample’s top-1 retrieved prototype remains within the clean sample’s top-5 retrieved prototypes (Top-5 Consistency), and (ii) the cosine similarity between the clean and corrupted top-1 prototype vectors. We observe that both metrics remain close to 100% under small perturbations, indicating highly stable retrieval. Even under strong noise (σ=0.30) or large occlusions (20×20, ≈39% area), the cosine similarity remains high (≈70% to 90%), demonstrating that retrieval either returns the same prototype or a semantically adjacent one.
> >
> > This behavior is inconsistent with harmful metastable states: small input changes do not trigger abrupt jumps to unrelated prototypes, which would indicate unstable attractor dynamics. Instead, a whole neighborhood of similar inputs maps to a compact region in prototype space, reflecting smooth and reliable retrieval trajectories. This also shows that V-HMN can discriminate between very similar patterns—perturbed variants of the same image consistently retrieve nearby prototypes rather than collapsing into distant classes. Together, these results confirm that the associative-memory mechanism supports stable pattern completion rather than problematic metastability.
> >
> > **If the number of parameters are about the same, is the main advantage of the model the improved accuracy (for which we should probably look at more datasets) or the amount of training data required. If more data is diluting the system due to the metastable states, then this may pose a limitation for this model which should be discussed.**
> >
> > Our model exhibits advantages in accuracy, data efficiency, and interpretability. The associative memory not only improves performance in low-data regimes by enabling effective prototype reuse, but also provides interpretability by allowing us to inspect which stored features are retrieved and how they influence the latent state and prediction, a transparent correction pathway that conventional feedforward backbones do not offer.
> >
> > Regarding the concern that increasing the amount of data might “dilute’’ the memory or create metastable states: although our memory stores real sample features, these features are produced by a jointly trained encoder and evolve throughout training. Thus, the memory forms a dynamic learned feature space rather than a fixed attractor set as in classical Hopfield networks. Empirically, we observe no degradation when scaling up the amount of data, including on ImageNet (Please refer to our new experiments on ImageNet in lines 426-454 and Table 6, in the revised manuscript), where refinement remains stable and beneficial, directly contradicting the dilution concern.
> >
> > **Is the code for this being made available?**
> >
> > Yes. The code will be released with the final version of the paper.

---

### Official Review · Reviewer_e5Z8 · 2025-10-30

**Soundness:** 3
**Presentation:** 3
**Contribution:** 2
**Rating:** 4
**Confidence:** 4

**Summary:**

This paper proposes an application of modern Hopfield networks (MHNs) to the domain of vision and classification.
Particularly, the application of MHNs to this domain enhances the known advantages of MHNs, like interpretability, data efficiency and biological plausibility. The proposed V-MHN architecture is extensively compared against existing architectures.
Empirically, V-HMN achieves strong results on data efficiency and demonstrates that memory-centric design can be a viable alternative to self-attention–based architectures in low data regimes.

**Strengths:**

The paper presents a novel architecture that is memory centric, which is a very interesting design philosophy that should be studied more in the design of interpretable neural networks.

Empirically, this architecture is competitive with strong prior baselines. Specifically, it outperforms prior work on data efficiency. The paper is well motivated.

Interpretability through explicit memory retrieval. The visualizations in Figure 2 are insightful and show that retrieved prototypes align well with semantic structure.

**Weaknesses:**

While the empirical results look strong for data efficiency, it is unclear why this is the case. An analysis or a dicsussion section on why this architecture is more data efficient than prior work would be very helpful.

Tables 3 and 4 are steps in the right direction to aid this understanding, but it appears that neither memory size nor number of steps on iterative refinement have an effect on the performance. The natural question that arises in that case is why is this architecture better than prior work.

The predictive-coding analogy is appealing but somewhat superficial. The update rule is effectively a weighted average with a learned coefficient. It is also unclear why the number of iterations is relevant if the coefficient is learned. The connection to biological predictive coding should be tempered or better substantiated.

The interpretability claim could be better quantified, although this is a minor weakness, as this is a direct consequence of the hopfield architecture.

Minor comments:  efficiency is misspelled as efficency in the abstract.

**Questions:**

On table 3:
1) Does zero iterations mean that iterative refinement is not done? In this case, is the memory bank unused? Why does the lack of a iterative refinement only marginally hurt the results?
2) Do you fix $\beta$ for the purpose of this part? if not, for iterations 1,2,3, wont the learning process of $\beta$ affect the results?
3) Would it be informative to repeat Table 3 using a small data fraction (e.g., 10\%) to highlight whether iterative refinement contributes most under data scarcity?

On table 4:
4. The results seem largely insensitive to memory size. Could you report retrieval hit-rate or memory-slot utilization to show that the memory is actively used?
5. I wonder how performance would change due to class imbalance?

---

> ### Author Response · Authors · 2025-11-25
>
> We are grateful to the reviewer for the insightful comment.
>
> # Weaknesses:
> **While the empirical results look strong for data efficiency, it is unclear why this is the case. An analysis or a dicsussion section on why this architecture is more data efficient than prior work would be very helpful.**
>
> We have added a new discussion section in the revision (see Section A.4), where we analyze the architectural components responsible for the improved data efficiency. We hope this clarifies the underlying reasons for the model’s strong performance for data efficient training.
>
> In addition, we also supplemented an experiment about robustness of V-HMN, please refer to Section A.12 lines 1167-1171 and its related Figure 6. The high stability of the Orange line indicates a tight prototype neighborhood. Vast noisy variants are mapped to a few semantic centers. This "many-to-one" robust mapping enables learning core concepts without seeing every variant, significantly enhancing data efficiency.
>
> **Tables 3 and 4 are steps in the right direction to aid this understanding, but it appears that neither memory size nor number of steps on iterative refinement have an effect on the performance. The natural question that arises in that case is why is this architecture better than prior work.**
>
> We agree that the effect of refinement iterations and memory size is not very significant when evaluated solely through clean-data top-1 accuracy. However, the revised manuscript now provides a more complete picture that clarifies why both components matter.
>
> First, Table 3 (Iterative Refinement: lines 340-352) shows that enabling even a single refinement step already yields consistent improvements across all datasets, with performance peaking at two steps. While the backbone is strong enough that accuracy does not collapse at t=0, the associative update evidently provides non-trivial gains.
>
> Second, Figure 2 (Robustness Experiments: lines 353-363) further demonstrates that refinement steps offer clear and consistent benefits under distribution shifts. Across Gaussian noise, occlusion, and contrast corruptions, accuracy improves steadily from t=0  → t=1→ t=2, with particularly pronounced gains for occlusion and contrast. These results highlight that iterative error-correction contributes meaningful robustness that is not captured by clean accuracy alone.
>
> Third, Table 4 (Memory Size Ablation: lines 365-377) shows that performance is highest with moderate memory sizes rather than very large ones. This indicates that the role of memory is not to maximize capacity, but to provide a well-curated prototype prior for rapid correction within one or two refinement steps. Too-small memories limit coverage, while overly large ones introduce redundancy and retrieval noise.
>
> Taken together,  V-HMN benefits from a synergistic interaction between a strong feedforward pathway and a lightweight, prototype-based prior that enhances robustness and data efficiency. The architecture is not merely increasing depth or width, but introducing a distinct computational mechanism, explicit associative retrieval and error-corrective refinement, that differentiates it from prior work.
>
> **The predictive-coding analogy is appealing but somewhat superficial. The update rule is effectively a weighted average with a learned coefficient. It is also unclear why the number of iterations is relevant if the coefficient is learned. The connection to biological predictive coding should be tempered or better substantiated.**
>
>  We agree that the predictive-coding (PC) connection should not be interpreted as a full biological implementation. In the revision, we have clarified that our formulation is PC-inspired (see Sec. A.2 Iterative Refinement as PC Dynamics). Regarding the update rule: although the refinement step can be written as a weighted combination with a learned coefficient, this does not reduce the process to a trivial averaging operation. The learned coefficient functions as a step size controlling how strongly the latent state incorporates the retrieved prototype; iterative application of this update leads to a nontrivial trajectory in latent space. Even with a learned coefficient, the number of refinement iterations remains relevant, because the refinement dynamics are not equivalent to a single-step closed form: the memory retrieval at each iteration is conditioned on the current latent state, meaning that successive iterations adjust both the query and the retrieved prototype. As a result, the refinement path is genuinely iterative rather than a single weighted average.

---

> ### Author Response · Authors · 2025-11-25
>
> **The interpretability claim could be better quantified, although this is a minor weakness, as this is a direct consequence of the hopfield architecture.**
>
> We agree that the interpretability claim can be strengthened.  To address this, we have added two  analyses in Section A.10. and Section A.11.
>
> 1. Memory-weight visualization (Figure 5). We visualize the average memory weights for each class and observe that inputs strongly activate prototypes from their own class (visible for CIFAR-10 and Fashion-MNIST). This provides a direct, interpretable view of how the model retrieves and uses stored memories.
>
> 2. Retrieval hit-rate analysis (Table 9).
> We compute the top-1 and top-5 class-consistency of the memory slots receiving the highest refinement weight. For CIFAR-10/100 and Fashion-MNIST, the hit rates are far above the random baseline (e.g., 10% → 76% top-1 on CIFAR-10), confirming that refinement consistently pulls representations toward prototypes of the correct class.
>
> **Minor comments: efficiency is misspelled as “efficency” in the abstract.**
>
> We have fixed this typo. Thanks.
>
> # Questions:
>
> **Does zero iterations mean that iterative refinement is not done? In this case, is the memory bank unused? Why does the lack of a iterative refinement only marginally hurt the results?**
>
> Yes. When t=0, the retrieval-and-update loop is skipped and the memory bank is allocated but not queried. As shown in Table 3, turning off refinement consistently reduces accuracy across all datasets. Although the numerical drop may seem moderate, gains of 0.5–1.2 percentage points are considered substantial on saturated small-scale benchmarks, where improvements beyond 0.3–0.5 points are typically meaningful. This behavior is expected: the feedforward path in V-HMN is already a strong standalone backbone, and the memory is designed to provide targeted error correction rather than full feature reconstruction. In predictive-coding–inspired architectures, the first correction step typically contributes most of the benefit, which explains why accuracy drops when refinement is removed but does not collapse. We have added a description of this behavior to lines 340–352 in the revised manuscript.
>
> **Do you fix β for the purpose of this part? if not, for iterations 1,2,3, wont the learning process of β affect the results?**
>
> Our goal in this ablation is to evaluate the overall effectiveness of different refinement depths. Each configuration (0, 1, 2, or 3 refinement steps) is therefore trained as an entirely separate model, and the refinement-related parameter, namely β, is optimized jointly within that configuration. This design is intentional: the refinement dynamics (step size and number of steps) are meant to co-adapt during training. As a result, the learning process of β  does not interfere across conditions, and the comparison reflects the best achievable performance for each refinement depth.
>
> **Would it be informative to repeat Table 3 using a small data fraction (e.g., 10%) to highlight whether iterative refinement contributes most under data scarcity?**
>
> We conducted an additional ablation using an even more extreme small-data setting (5% of the training data), and varied the number of refinement steps $t\in{0,1,2,3}$ accordingly. The results are shown in the following table.
>
> | Iterations | CIFAR-10 | CIFAR-100 | FashionMNIST |
> |-----------:|----------:|-----------:|--------------:|
> | 0 | 70.69 | 31.02 | 87.16 |
> | 1 | 71.77 | 31.84 | 87.62 |
> | 2 | 71.78 | 31.87 | 87.64 |
> | 3 | 71.73 | 31.69 | 87.64 |
>
> Across all datasets, we observe the same qualitative trend as in the full-data setting. These results show that the relative benefit of refinement is also present under data scarcity, and the optimal number of refinement steps remains small, consistent with the predictive-coding interpretation that one or two iterations suffice to correct most inputs even when data is limited.
>
> **On table 4: 4. The results seem largely insensitive to memory size. Could you report retrieval hit-rate or memory-slot utilization to show that the memory is actively used?**
>
> In order to show whether memory is actively used, we report the top-1 and top-5 retrieval hit-rate. please refer to the Section A.11 and Table 9.

---

> > ### Author Response · Authors · 2025-11-25
> >
> > **I wonder how performance would change due to class imbalance?**
> >
> > We conducted additional experiments on imbalanced CIFAR-10/100 with imbalance ratios of 50 and 100.  Please refer to Section A.9 and Table 8. Across all settings, V-HMN exhibits the strongest robustness to class imbalance，We attribute this behavior to the associative-memory mechanism:
> >
> > Unlike purely feedforward models, V-HMN maintains learned prototypes that aggregate information across the entire training distribution. During inference, the refinement step retrieves class-relevant prototypes and pulls the latent representation toward them. Even under severe imbalance, minority-class prototypes remain available in the memory bank, providing a stabilizing signal that mitigates representation drift and reduces majority-class dominance. These results indicate that associative-memory refinement acts as an inherent regularizer in long-tailed regimes, enabling V-HMN to maintain accuracy where other architectures degrade more severely.

---

### Official Review · Reviewer_FEDf · 2025-11-02

**Soundness:** 3
**Presentation:** 3
**Contribution:** 3
**Rating:** 8
**Confidence:** 4

**Summary:**

The paper proposes a Vision Hopfield Memory Network (V-HMN), a biologically inspired foundation model that replaces self-attention and convolutional mixing with local and global Hopfield memory modules. Each block retrieves prototypes from memory banks (class-balanced, written during training) and performs iterative refinement akin to predictive-coding error correction. V-HMN achieves competitive results on CIFAR-10/100, SVHN, and Fashion-MNIST, with claimed advantages in data efficiency and interpretability.

**Strengths:**

- The work demonstrates the benefit of a memory based system in perceptual learning, bridging the gap between memory / prototype based system and distributed system. This shows a better alignment with brain theory and practically improves the data efficiency and interpretability in perceptual models.
- The work shows comprehensive ablation studying in various functions and roles of hyperparameters.
- Presentation is clear and concise.

**Weaknesses:**

- All benchmarks are small-scale (≤ 32×32 images). Claims about “foundation backbone” or “multimodal generalizability” are not validated on large datasets (ImageNet, ADE20K, etc.). Maybe add a small data with slightly large data resolution.
- The idea of using memory banks to improve the data efficiency is not new (E.g. [1] in few shot image generation – to test the limits of data efficiency). The author should consider a more comprehensive background review in terms of the computational benefits of the prototype memory.
- The paper only demonstrates the benefits of the memory based system on classification but it would be more beneficial to see if the method would generalize to other types of tasks. But it’s a promising direction that should be shared with the community.
[1] Li, Tianqin, et al. "Prototype memory and attention mechanisms for few shot image generation." International Conference on Learning Representations. 2022.

**Questions:**

For the iterative refinement, does it actually change different retrieved prototype in each iteration? Whether it will actually help to retrieve more related prototypes (i.e. broader sense of pattern completion to bring stronger prior to it).

---

> ### Author Response · Authors · 2025-11-25
>
> We thank the reviewer for the positive and insightful comments.
>
> # Weaknesses:
>
> **All benchmarks are small-scale (≤ 32×32 images). Claims about “foundation backbone” or “multimodal generalizability” are not validated on large datasets (ImageNet, ADE20K, etc.). Maybe add a small data with slightly large data resolution.**
>
> We have now included ImageNet-1k experiments, please kindly refer to lines 426-455 and Table 6. V-HMN remains competitive on ImageNet, indicating that the framework is not only biologically inspired but also inherently interpretable and sufficiently general to operate effectively at high-resolution, large-dataset regimes. Plus, V-HMN preserves a transparent computational structure, namely explicit associative-memory retrieval coupled with predictive-coding–inspired iterative refinement, that is difficult to maintain in more complex or heavily engineered architectures. This transparency further enables interpretability of the decision process, since predictions can be explained by identifying which stored patterns are retrieved and refined.
>
> **The idea of using memory banks to improve the data efficiency is not new (E.g. [1] in few shot image generation – to test the limits of data efficiency). The author should consider a more comprehensive background review in terms of the computational benefits of the prototype memory.**
>
> We have added related work, and please refer to Section 2.2 (lines 123-135). If there are further relevant works the reviewer thinks should be included beyond those added in our revision, we would be glad to incorporate them in the final revision.
>
> **The paper only demonstrates the benefits of the memory based system on classification but it would be more beneficial to see if the method would generalize to other types of tasks. But it’s a promising direction that should be shared with the community. [1] Li, Tianqin, et al. "Prototype memory and attention mechanisms for few shot image generation." International Conference on Learning Representations. 2022.**
>
> The current submission focuses on classification due to space and computational limits, and thus does not fully showcase the broader applicability of the memory mechanism. In the revised version, we have added a short discussion on how V-HMN can naturally extend to other tasks, such as retrieval, few-shot learning, or simple downstream reasoning, and we kindly refer the reviewer to that section 5 (lines 525-538). We also agree that the proposed memory architecture is task-agnostic and offers promising opportunities for future exploration beyond classification.

---

> > ### Author Response · Authors · 2025-11-25
> >
> > # Questions:
> >
> > **For the iterative refinement, does it actually change different retrieved prototype in each iteration? Whether it will actually help to retrieve more related prototypes (i.e. broader sense of pattern completion to bring stronger prior to it).**
> >
> > To address the reviewer’s question regarding whether iterative refinement retrieves different prototypes across iterations, and whether it strengthens class-consistent priors (i.e., broader pattern completion), we performed a dedicated analysis of retrieval dynamics in the last HMN Block, and the results are shown in the below table:
> >
> > | Iterations | Mean cosine similarity (before) | Mean cosine similarity (after) | GT attention mass (before) | GT attention mass (after) |
> > |-----------:|--------------------------------:|-------------------------------:|----------------------------:|---------------------------:|
> > | 1 | 0.929 | 0.583 | 41.82% | 55.74% |
> > | 2 | 0.955 | 0.352 | 25.94% | 49.71% |
> > | 3 | 0.970 | 0.298 | 13.71% | 31.97% |
> >
> > Here, *“GT attention mass”* measures how much attention weight is assigned to prototypes belonging to the ground-truth class. Formally, it is the sum of attention weights over all prototypes whose stored class label matches the input. Higher values indicate that the retrieval distribution places more probability mass on class-consistent prototypes. *“Mean cosine similarity”* compares the latent representation to the initial memory readout and reflects how far refinement moves the representation in prototype space. *“Before”* refers to the state at the first memory readout (t = 0), before applying any refinement updates. *“After”* refers to the final state after performing t iterations of refinement (t ∈ {1, 2, 3}).
> >
> >
> > We can observe two clear phenomena:
> >
> > 1. The retrieved prototypes do change across iterations. Cosine similarity provides a direct measure of how much the final latent state deviates from the initial prototype readout. If the refinement loop were not retrieving different prototypes, the final vector would remain aligned with the initial memory readout, resulting in high cosine similarity. Instead, we observe a sharp drop (0.583→0.298 from t=1→3), indicating that the representation moves toward different regions of prototype space across iterations. This demonstrates that iterative refinement indeed retrieves different，and increasingly more class-consistent prototypes.
> >
> > 2. Refinement consistently increases GT attention mass, indicating stronger class-relevant priors.
> > GT attention mass increases after refinement for every t:
> >
> > t=1: 41.82% → 55.74%
> >
> > t=2: 25.94% → 49.71%
> >
> > t=3: 13.71% → 31.97%
> >
> > This means that as the model iterates, a larger fraction of attention is assigned to prototypes from the correct class, which is exactly the “broader pattern completion” behavior. As refinement proceeds, a larger fraction of the attention distribution shifts toward prototypes belonging to the correct class, indicating that the model increasingly retrieves class-consistent prototype neighborhoods rather than a single fixed prototype.
> >
> > Together, the decrease in cosine similarity (indicating movement to new prototypes) and the increase in GT attention mass (indicating stronger class-consistent prior) demonstrate that:
> >
> >  (i) each iteration retrieves a different, progressively more relevant prototype, and
> >
> >  (ii) refinement amplifies class-aligned memories, helping the model converge toward semantically consistent prototypes even when the initial retrieval is noisy.
> >
> > These findings show that iterative refinement is not redundant but plays a key role in guiding the system toward more meaningful and class-consistent prototype regions.

---

### Author Response · Authors · 2025-11-25
**To All Reviewers**

We sincerely thank all reviewers for their thoughtful and constructive feedback. Across the revision, we incorporated several substantial improvements addressing common concerns raised throughout the reviews. We clarified the conceptual positioning of our method, strengthened the discussion of predictive-coding–inspired and Hopfield-style refinement dynamics, and expanded the analysis explaining V-HMN’s data efficiency and robustness. We also added new experiments, including ImageNet-1k results, small-data and imbalance studies, iterative-refinement analysis, memory-usage analyses, and stability evaluations under various corruptions.

In addition, we would like to note that the ImageNet-1k experiments were conducted under limited time and GPU resources, and we were not able to perform extensive hyperparameter search or architecture-specific tuning. As V-HMN is a newly proposed architecture, its training configuration has not yet undergone the level of refinement that mature CNN and Transformer families have benefited from over years of development. For these reasons, the current ImageNet results should be viewed as a preliminary baseline rather than a fully optimized performance level, and we expect that further tuning may reveal additional improvements. Even so, obtaining accuracy comparable to widely used backbones under minimal tuning is encouraging and suggests that the core inductive bias of V-HMN remains effective at large scale.

We also improved the interpretability analyses, added missing related work, corrected minor issues, and refined wording for clarity. All new or updated content is highlighted in blue in the revised manuscript.

Finally, we would like to emphasize that beyond competitive accuracy, V-HMN provides additional benefits in **data efficiency**, **interpretability**, and its **brain-inspired computational perspective**. Its associative memory enables effective prototype reuse in low-data regimes and offers a transparent way to inspect which stored features are retrieved and how they influence the prediction, an aspect that conventional feedforward backbones typically lack. Moreover, because the model is built upon explicit pattern storage and retrieval, this computational principle naturally extends to tasks such as retrieval-oriented learning, few-shot recognition, dense prediction, and domain adaptation, where explicit prototypes can offer an advantageous inductive bias. Furthermore, the memory module opens the possibility for online test-time adaptation: by incorporating unlabeled test representations during inference, V-HMN can gradually adjust its prototype landscape, which may benefit scenarios that require continual interaction with dynamic environments, such as robotics or autonomous driving.

Below, we provide point-by-point responses to each reviewer.

---

### Note · Program_Chairs · 2026-01-17
**Submission Desk Rejected by Program Chairs**

The following references in this submission do not refer to real documents and/or have major errors in bibliographic information:

 Siyuan Qiao, Liang-Chieh Chen, Bo Yang, Alan Yuille, and Serge Belongie. Prototype memory and attention mechanisms for few-shot image generation. In International Conference on Learning Representations, 2022.